# Automatic authorship attribution in Albanian texts

**Arta Misini** [1,2]*, **Ercan Canhasi** [1]*, **Arbana Kadriu** [2], **Endrit Fetahi** [1]*

**1** Faculty of Computer Science, University of Prizren, Prizren, Kosova, **2** Faculty of Contemporary Sciences and Technologies, South East European University, Tetovo, North Macedonia

* arta.misini@uni-prizren.com (AM); ercan.canhasi@uni-prizren.com (EC); endrit.fetahi@uni-prizren.com (EF)

**Data Availability Statement:** All data files are available from the Zenodo database (accession number(s) 10.5281/zenodo.12699563.); link: https://zenodo.org/records/12699563.

**Funding:** The author(s) received no specific funding for this work.

## Abstract

Automatic authorship identification is a challenging task that has been the focus of extensive research in natural language processing. Regardless of the progress made in attributing authorship, the need for corpora in under-resourced languages impedes advancing and examining present methods. To address this gap, we investigate the problem of authorship attribution in Albanian. We introduce a newly compiled corpus of Albanian newsroom columns and literary works and analyze machine-learning methods for detecting authorship. We create a set of hand-crafted features targeting various categories (lexical, morphological, and structural) relevant to Albanian and experiment with multiple classifiers using two different multiclass classification strategies. Furthermore, we compare our results to those obtained using deep learning models. Our investigation focuses on identifying the best combination of features and classification methods. The results reveal that lexical features are the most effective set of linguistic features, significantly improving the performance of various algorithms in the authorship attribution task. Among the machine learning algorithms evaluated, XGBoost demonstrated the best overall performance, achieving an F1 score of 0.982 on literary works and 0.905 on newsroom columns. Additionally, deep learning models such as fastText and BERT-multilingual showed promising results, highlighting their potential applicability in specific scenarios in Albanian writings. These findings contribute to the understanding of effective methods for authorship attribution in low-resource languages and provide a robust framework for future research in this area. The careful analysis of the different scenarios and the conclusions drawn from the results provide valuable insights into the potential and limitations of the methods and highlight the challenges in detecting authorship in Albanian. Promising results are reported, with implications for improving the methods used in Albanian authorship attribution. This study provides a valuable resource for future research and a reference for researchers in this domain.

## 1 Introduction

Authorship attribution (AA), identifying the author of a given text based on writing style, is a longstanding and essential problem in natural language processing (NLP). It is a crucial

**Competing interests:** The authors have declared that no competing interests exist.

problem in various fields, including forensics [1–5], journalism [1, 6–12], and literature [13–20], where determining authorship can have legal, social, and cultural implications.

The author's identification behind an anonymous text has a rich history dating back to the 19th century. Early studies focused on analyzing the writing styles of well-known authors such as Shakespeare [21] and the Federalist Papers [22, 23] using statistical techniques. To attribute authorship of written texts, researchers use a set of stylistic features that are unique to an individual's writing style, such as the use of punctuation [12, 24, 25], word choice [10, 11] and sentence structure [26]. It is possible to determine the author of an anonymous text by analyzing these features.

Three main components are typically needed to perform AA: a dataset of texts written by various authors, a set of stylistic features extracted from these texts, and a classification model that uses these features to identify the authors of anonymous texts. The dataset should be carefully selected to represent the characteristics of the authors being analyzed. Stylometric features are markers of writing style that can distinguish documents written by different authors and can be extracted from the dataset using profile-based, which combines an author's works into a single file and analyzes it, or instance-based, which examines each author's documents individually. A classification model then uses these extracted features to determine the true author of an anonymous text. There are various techniques and algorithms available for this purpose.

With the proliferation of digital text and the increasing importance of identifying the authors of online content, the demand for accurate AA methods has grown in recent years. Previous research on AA has primarily focused on languages such as English [2–5, 8, 10, 15, 18, 25–28], Spanish [29, 30], and Chinese [16, 20], with relatively little attention given to less studied languages. Therefore, the need for large, high-quality annotated corpora in many languages (Urdu [11, 12], Bengali [7, 19, 31], Arabic [32], and Turkish [6]), including Albanian [13], often impedes researchers in developing effective AA models.

## 1.1 Aims of the research

Albanian is a member of the Indo-European language family but is categorized as a separate subgroup. It is a unique and understudied language with a rich literary tradition and a growing body of online content. However, to the best of our knowledge, there currently needs to be a corpus designed explicitly for the study of AA in Albanian.

This paper presents the first-ever reference corpus of authorship attribution in Albanian, explicitly compiled to develop and evaluate AA models. The corpus consisted of newsroom columns and literary works and was created through a combination of web scraping and manual book scanning. We describe the corpus creation process, including the selection of texts, the annotation process, and the data collection approach. We also conduct experiments for AA on this corpus, using a range of features and machine learning (ML) techniques, and discuss the implications of our findings for future research in this area.

By newsroom columns, we refer to the written content produced by journalists and editors in a newsroom, including news articles, opinion pieces, analysis, and other forms of editorial content. While literary works are written works considered to have artistic or literary merit, which can include various forms of fiction, poetry, drama, and non-fiction, examples of literary works include novels, poems, short stories, and essays.

## 1.2 Research questions

Given our examination, we have carefully crafted several research questions to further investigate the problem at hand. These research questions are as follows:

**RQ** *1: What are the most effective linguistic features for accurately identifying the authors in Albanian texts?*

**RQ** *2: Can machine learning approaches be effectively used for authorship attribution in Albanian language texts, and if so, which algorithms perform the best?*

**RQ** *3: What are the effects of multiclass classification strategies on authorship attribution?*

## 1.3 Contributions of the paper

This study has contributed significantly in the following ways. First, we provide a novel resource for studying AA in Albanian, which can be a reference for future research in this field. Second, we demonstrate the usefulness of this corpus in developing and evaluating AA models. Our work aims to advance the understanding of AA in Albanian and facilitate the development of more effective methods for this problem.

## 1.4 Organization of the paper

The remaining sections of this paper are organized as follows. The next section provides an overview of the existing research on AA. Section 3 compiles the first-ever extended Albanian authorship attribution corpus (A3C-e). In Sections 4 and 5, we report our experiments and the resulting findings, addressing the research questions. Finally, we conclude with a summary of our findings and suggestions for future work in Section 6.

## 2 Related work

This section aims to provide a comprehensive overview of the research conducted on authorship attribution in different languages. We will focus on three main components: feature extraction, classification methods, and AA corpora.

## 2.1 Feature extraction

One of the critical challenges in AA is the extraction of features and patterns that can be used to distinguish the writing styles of different authors. Researchers have used a variety of linguistic traits for the AA problem, including lexical, morphological, syntactic, semantic, and structural features.

Lexical features [17, 25] include the use of specific characters, words, and n-grams that capture the distinctive word usage patterns in an author's writing. The text can be analyzed at the character level and word level. These features include some basic measures that can be calculated for any text, such as word length [7, 9, 17], frequency of specific words [25], and the number of words per sentence [19]. Khatun et al. [33] claimed that character-level features (the number of characters [9, 11, 12, 17, 34], letters [10, 12, 27, 34], the use of digits [10–12, 34], and white spaces [11, 12, 34]) are sufficient for AA. Additionally, techniques such as co-occurrence of word pairs [19], vocabulary richness [10–12, 24, 34], and n-gram analysis [7, 9, 11, 12, 14, 32, 35, 36] provide further insight into the author's writing style.

Morphological features focus on the grammatical structure of a text and include part-of-speech (POS) tags and inflectional forms. Researchers have used POS tagging to identify the frequency of specific tags [18], such as nouns, verbs, and adjectives [29]. Additionally, function words [17, 27, 37], stop words [5, 10, 14, 24], and POS n-grams [3, 15, 27], have also been used as a feature, with the most frequent n-grams being selected for analysis. In addition to POS tagging, morphological features include inflectional forms, such as verb tense and word

pluralization [7]. Morphological features are an essential aspect of stylometry, as they have been used to identify patterns in how authors use language.

Structural features, such as the organization and structure of texts, have also been used in AA. This includes the analysis in sentence- [3, 19, 28, 34, 38], paragraph- [19, 28], and document-level [12] as reported in studies by Romanov et al. [25] and Hriez et al. [17]. Belvisi et al. [2] and others have analyzed features such as greetings and farewells in emails and the average length of paragraphs and sentences [27]. Islam et al. [7] also analyzed the length of words and sentences.

Syntactic and semantic features [9] can be used together to provide a comprehensive understanding of an author's language and writing style that captures the syntactic and semantic structure of texts [17].

Syntactic features focus on the syntactic structure [25, 26] of a text, including elements such as syntactic phrases [9, 36], relations [27], and writing rules [6, 9]. Wu et al. [10] introduced syntactic structures and dependencies [10, 27] as additional features in the feature extraction process, in addition to words. Researchers can use these features to identify patterns in how authors construct sentences.

Semantic features, on the other hand, require a deeper level of linguistic analysis focusing on the meaning of the text. Researchers have employed active and passive voice verbs [17] in a text to classify authors. On the other hand, Hossain et al. [19] used semantic similarity measures on word pairs. However, the authors noted that the document-level similarity between semantics and syntactic could also contribute to misclassification in AA. The reliability of semantic analysis may need to be higher in under-resourced languages, leading to limited use of high-level semantic features in stylometry, particularly in AA.

Stylometry, the study of linguistic style, uses stylometric features to analyze and identify an author's writing style. These features, such as readability measures [12, 26], the use of capitalization [3], punctuation [9, 12, 24, 25, 27, 28, 36], sentence structure [25, 26] and word choice [10–12, 27], are statistical properties of the author's writing style. Combining different feature categories has been shown to improve the performance of AA, as demonstrated by Hriez et al. [17], Wu et al. [10], and Ramezani [9].

Researchers use these features to study writing styles and identify author-specific language characteristics. AA techniques have been extended in various languages by the use of ML-based algorithms for text analysis.

## 2.2 Classification methods

In recent years, there has been a growing interest in developing more effective methods for AA. Several studies have explored the use of these methods for AA in different languages.

Researchers have focused on developing automatic methods for AA through the effectiveness of ML-based techniques in automated classification, using various algorithms to classify text samples based on authorship.

Some popular ML-based algorithms used in AA include Support Vector Machines (SVMs) [6, 32, 36], Bayesian classifiers [12, 18], k-Nearest Neighbors (k-NN) [11, 34], Random Forests (RF) [16], Stochastic Gradient Descent (SGD) [19], Tree-based classifiers [14, 16, 17, 28], and Logistic Regression (LR) [10, 12, 18, 35]. In most studies, the authors used a combination of these algorithms to classify the text. A diverse set of features and different classifiers with ensemble learning methods [4, 29, 38] has been used to classify authorship. Other ensemble techniques, such as gradient boosting and a voting classifier, were used by Singh et al. [27] and Taha et al. [39]. In this way, Misini et al. [40] have presented various techniques used in authorship-related tasks in different languages.

In [2], researchers use a new technique to short-text authorship detection. They calculated the similarity of n-gram vectors using three different distance metrics [3]. Furthermore, recent work includes deep learning (DL) techniques for AA.

DL techniques, such as convolutional neural networks (CNNs) [12, 19, 33], recurrent neural networks (RNNs) [41] (long short-term memory (LSTM) networks [8, 15, 25], gated recurrent unit (GRU) [8]), have been used to analyze text data and identify authors based on their writing style. A different model for author classification has been proposed recently. A self-attention mechanism is employed in [10, 26] to model word dependencies in texts, improving model performance when combined with CNNs or transformers [20, 25].

Pre-trained language models [42] have become essential in NLP, providing significant improvements in various tasks. Prominent models such as BERT [43, 44], RoBERTa [45], XML-RoBERTa [46] have set new models for performance and efficiency in NLP applications.

In AA, word embeddings have been used in ML models to identify the author of a text. The distinct distribution of words a specific author uses in the vector space allows the model to distinguish the author's writing style. Popular word embedding models include word2vec [19, 41], GloVe [12, 19, 41], Skip-gram model [33, 41], and fastText [12, 19, 33, 41]. Researchers actively study AA, and the field continues to evolve with new methods and approaches.

## 2.3 Authorship attribution corpora

One of the key elements in research on authorship attribution is using the dataset, or collections of texts, to evaluate the performance of different techniques. A variety of corpora have been used for AA in other languages, including English [30, 37, 47, 48], Spanish [30, 48], French [48], Chinese [20] among others. However, there is a scarcity of annotated corpora for under-resourced languages, such as Urdu [11, 12], Bengali [19, 24, 31], Arabic [17, 32], Albanian [13], and Turkish [6]. The use of appropriate corpora is crucial for AA research, as the corpora characteristics and features extracted from the dataset can influence the study results.

Our focus in this study is on corpora that consist of either newsroom columns or literary works. These are closely aligned with the composition of our novel corpus, providing a relevant and applicable context for our research. Later in this section, we will also consider other types of documents, further broadening our understanding of AA across different genres and styles.

To provide an overview of the research on AA in different languages, we have compiled a table of corpora used in previous studies (Table 1). The table includes information on the language, the types of texts (literary or columns), and the corpus size. This table highlights the diversity of corpora available for this purpose.

As demonstrated in Table 1, several studies have used different corpora for AA in various languages. These corpora have been used to evaluate the performance of AA techniques. However, the need for more standard datasets, especially for low-resource languages, makes AA challenging. Corpora, such as UNAAC-20 [12] in Urdu, and BAAD16 [19, 31, 33] in Bengali, were created using a semi-automatic approach of manual validation and automated scraping, and are significant contributions to the field.

In the field of AA, researchers have used a variety of corpora (collections of texts) to study the writing styles of different authors. These corpora consist of different types of documents, including literary works [13, 17, 18, 20], newsroom columns [1, 7, 12], tweets [3, 32], blog posts [25, 26], and other short written texts [5, 10, 27, 34, 36].

Researchers have developed several corpora specifically for performing AA on literary works, such as novels [15, 16], poems [20], and other written works [28]. Some common features include lexical features [15, 17, 31] (character- [16, 20] and word-level [28]), n-grams

**Table 1. Authorship attribution corpora in different languages.**

| Reference | Dataset | Language | Number of authors | Number of samples |
|---|---|---|---|---|
| | **Newsroom columns** | | | |
| [37, 47] | TREC corpus | English | 7 authors | 5,600 documents |
| [26, 49–51] | Reuters Corpus Volume 1 | English | 2,361 authors | 109,433 texts |
| [35, 50] | Guardian corpus | English | 13 authors | 444 documents |
| [6] | Kibris dataset | Turkish* | 7 authors | 50 columns per author |
| [12] | UNAAC-20 | Urdu* | 94 authors | 21,938 columns |
| | **Literary works** | | | |
| [52] | German Corpus | German | 2 authors | 3 novels per author |
| [19, 31, 33] | BAAD16 | Bengali* | 16 authors | 17,966 sample texts |
| [17] | AAAT dataset | Arabic* | 7 authors | 7 books (71 paragraphs) |
| [18] | Gungor 50 | English | 45 authors | 25,636 records |
| [20] | Classical poetry corpus | Chinese | 20 poets | 11,289 poems |

* stands for low-resource languages

[15, 17], POS tags [15, 18, 28], syntactic features [15, 17], and structural features [17, 28]. Also, the use of function words [17], rhythmic features [16, 17], and topic modeling [20] gave favorable results. ML-based algorithms used include SVM [16, 18, 28], Decision Trees (DT) [16, 17, 28], and Neural Networks (NN) [24, 41] (CNN [15, 20], LSTM [15, 31]). The use of transformers using multi-head attention [20, 31] also has been proposed to advance the AA research.

To perform AA on newsroom columns corpora, researchers have utilized techniques such as ML-based algorithms such as SVMs [12], k-NN [11], and LR [12]. Also, neural networks [7, 12] including CNNs [12] and RNNs (LSTM [8, 12, 31], GRU [8]) have been applied. Other techniques that have been employed include lexical analysis [1, 9, 10, 31], morphological analysis which involves POS tags [9], and stop words [10], and structural analysis [7, 12, 25].

In the AA domain, researchers have made progress in identifying writing styles and attributing anonymous texts, but there are still challenges to be addressed. One key challenge is the limited information available for analysis, making it difficult to identify and analyze the linguistic patterns and structures indicative of authorship.

Albanian has received limited attention in previous research as a resource-poor language in terms of NLP. To address this challenge, we have developed a novel reference corpus for attributing authorship in Albanian. It is the first reference corpus for the Albanian language. The complete workflow used in building the corpus is described in the following section.

## 3 Albanian authorship attribution corpus

This section introduces the extended Albanian authorship attribution corpus (A3C-e), a newly created resource-poor language corpus explicitly developed specifically for AA research in Albanian. It comprises various text samples, including newsroom columns and literary works. It was done intentionally to provide a well-rounded representation of language usage in different contexts and evaluate the performance of AA techniques in various writing styles. Newsroom columns were selected for their rich vocabulary covering various field topics. They are typically publicly accessible without charge and with fewer usage restrictions. Moreover, they have a history of being used in AA corpora [53–55] and have been utilized in Albanian corpus building for text classification [56, 57] and detecting fake news [58]. The inclusion of literary works, on the other hand, allows for evaluating AA techniques on more sophisticated

language, further diversifying the corpus. Literary works [13] have the ability to exhibit a high degree of linguistic style and provide a representative sample of an author's writing, including various genres and themes. The authors often have a consistent writing style that can be used to distinguish their works.

## 3.1 Data collection methodology

Creating a corpus of text samples attributed to specific authors is critical to AA. When selecting a corpus, it's essential to consider corpus size, type of texts and authors included, and alignment with the research questions as discussed in [59, 60]. By collecting a large and diverse set of texts from various sources, the corpus enables to capture the full range of variation in language use and evaluate the performance of different AA techniques under various conditions.

The data collection methodology involved scraping multiple news websites to gather various columns. Book scanning was also carried out to collect literary works from the contemporary literature period.

**3.1.1 Web scraping for collecting newsroom columns.** Collecting newsroom columns from various websites for our A3C-e corpus was done through web scraping. With the help of web scraping, it is possible to gather massive volumes of textual data from the web efficiently. We developed a custom web scraper to extract information from websites automatically by sending HTTP requests and parsing the HTML response. The web scraping process involved a semi-automatic method that combined manual source searches with automated data extraction.

The first step in the web scraping process was identifying the websites to target. This was done by searching for online news portals that regularly published newsroom columns in Albanian. Multiple news websites were selected for data extraction to ensure a diverse set of columns. These sources were selected based on their prominence and the likelihood of them having columns written by single authors. Some of the main sources for data collection were "javanews.al", "shenja.tv", "telegrafi.com", "koha.net", "syri.net", and "portalb.mk".

The automatic part of the web scraper uses the Python library "selenium". This library enables data scraping from web pages and provides a simple and easy-to-use API.

The second step in the process was to extract a list of column links from selected online web portals. This was done to ensure that the data collected was unique and that duplicates were eliminated. The web scraper was programmed to extract this list of column links, which were then used as the input for the next step in the data collection process.

Once the list of column links was obtained, the web scraper was used to visit each of these links and scrape the content of each column. It was achieved by sending HTTP requests to the target websites to retrieve the HTML code for the relevant pages. The columns' content was extracted by parsing the retrieved HTML code. The extracted data, primarily text, is then stored in a structured format, such as a CSV file with UTF-8 encoding compatible with Albanian text, expanding the available data for AA analysis.

To ensure the completeness of the corpus, it was also necessary to collect meta-information about the columns, such as the title, author, and date, in addition to the full text. It is important to note that the online sources' meta-information (author, date, title) varied considerably. Some sources may only provide the title and date of the column, while others may include the author and full text. We took care to collect this information for each column, as it can be useful in future AA analysis.

The next step was to pre-process the scraped data to remove any irrelevant information and format the text properly for use in the corpus. It included removing the text's navigation links, HTML tags, and redundant white spaces. Finally, the pre-processed columns were manually

reviewed to ensure their relevance and quality and eliminate duplicates or irrelevant information. To ensure the validity and reliability of the corpus, columns that were behind a subscription or registration barrier, typically labeled as premium content, were excluded from the corpus. It provided a corpus of newsroom columns accessible to the general public, allowing for broader dissemination and use of the corpus in future research and analysis.

Using the custom web scraper greatly simplified the data collection process, reducing manual effort and increasing the efficiency of the data collection process. By automating the data collection process, a large amount of data was obtained in a relatively short amount of time. The result was a large and diverse corpus of Albanian newsroom columns suitable for AA research.

The web scraper developed for collecting newsroom columns was essential in creating the novel A3C-e corpus for experimentation and evaluation. This corpus will serve as a valuable resource for researchers in the field of NLP, particularly for those interested in AA in the Albanian language.

**3.1.2 Book scanning for literary works.** The book scanning process in the Albanian AA task was crucial for collecting the literary works by various authors. This process involved digitizing physical books by capturing images of their pages and converting the images into text using Optical Character Recognition (OCR) technology. This process aimed to create a representative corpus of literary works written in Albanian.

To obtain a comprehensive and diverse dataset, we chose to scan books from the University of Prizren Library, which has a rich collection of contemporary literary works in Albanian.

The first step in the book scanning process was to select the books to be digitized. It was done by reviewing the available books in the University of Prizren library and selecting those written by well-known authors from contemporary literature in Albanian. To ensure the best quality of our corpus, we decided to focus on authors with many books in the library. The authors selected from contemporary literature represented diverse styles and genres.

Once the authors were selected, we began the process of book scanning using the Viisan book scanner and its accompanying Cambook application. The Viisan book scanner is a high-quality device for digitizing large quantities of printed books and manuscripts. The Cambook application is a user-friendly software that allows for easy and efficient scanning of books. It utilizes a high-resolution camera to capture the pages of a book, which is then processed to produce digital images. The selected books were then placed on the Viisan book scanner and scanned in high-resolution grayscale mode, resulting in high-quality digital images of the pages.

Approaching the book scanning process with caution was necessary due to copyright considerations, and selected pages were scanned to avoid copyright infringement. A systematic approach was adopted to achieve this, where every fifth page of the book was scanned, alternating between scanned and unscanned pages. This method ensured that the content collected was limited under copyright laws while still allowing for the collection of a representative sample of the author's writing style.

The scanning process was carried out in time sequences over approximately one month, ensuring that all the selected books were scanned with high quality and accuracy. The scanned books were then saved as digital files, with each scanned page being saved as an image file.

As a result of the careful scanning process, high-quality text outputs were produced that accurately reflected the original content of the author's writing. The next step was to convert the image files into text format, which was then saved in a UTF-8 encoding compatible with Albanian text. This step was necessary to make the text data usable for AA analysis.

The scanned images were processed using OCR technology to convert the images into text. It involved analyzing the scanned images to recognize the text and convert it into machine-

readable form. The OCR process was done using the integrated software of the CamBook application designed to recognize text accurately in various languages, including Albanian. The resulting text outputs were saved in text files, allowing easy integration with our corpus.

To build a robust and comprehensive corpus for AA, we took great care to ensure the quality of the data collected through the book scanning process. The manual validation and improvement of the scanned files were critical in ensuring that the corpus was accurate, complete, and suitable for use in the AA task.

Once the OCR process was complete, the resulting text files were carefully reviewed to ensure high quality. The manual validation process involved checking the scanned pages for any missing text, incorrect OCR outputs, or other errors that might have occurred during the scanning process. Any such errors were corrected manually by rescanning the affected pages or manually editing the scanned files. Efforts were made to ensure the text files were well-formatted and free of any issues that could negatively impact their use in the AA task.

Finally, the extracted text was pre-processed to remove irrelevant information and properly format the text for use in the corpus. It included removing page numbers, footnotes, headers, and other non-textual elements and ensuring the text was formatted correctly and error-free. The pre-processed text was then used to create a comprehensive corpus of Albanian literary works that accurately represented the original authors' writing.

The resulting corpus of literary works serves as a valuable resource for research in Albanian AA, ensuring its availability for future study and analysis.

In this study, we used the Viisan book scanner and the Cambook application to digitize a collection of literary works from the University of Prizren library. Collecting and compiling a corpus of literary works for AA in Albanian required automatic and manual techniques. By carefully selecting authors, using high-quality scanning equipment, and performing manual validation and improvement, we created a corpus suitable for use in the AA task. We compiled our corpus using a hybrid methodology, combining automatic and manual steps to ensure reliability. Fig 1 illustrates the complete workflow used in the corpus building.

The icons included in the graphic illustration of corpus development for each dataset are obtained from the Noun Project.

This corpus will be a valuable resource for researchers working on Albanian AA. It will contribute to a better understanding of the challenges and opportunities of this task in this under-resourced language.

### 3.2 Specifications of A3C-e corpus

This section outlines the specifications of the A3C-e corpus, including its size and summary statistics for several NLP features, to understand the data and its potential uses better. The proposed A3C-e corpus is a comprehensive and diverse collection of newsroom columns and literary texts, and its specifications are detailed in Table 2.

This table presents three crucial levels of linguistic information about the corpus, including corpus size, authorship statistics, and POS tags. The first level provides a comprehensive overview of the size of the corpus, while the second level provides insights into the distribution of text data among different authors. Finally, the last level showcases the POS-related information in the corpus.

Despite previous work on POS tagging for the Albanian language [61–63], the POS-related information in this corpus was generated using an internally developed POS tagger that was trained on a large dataset of annotated texts in the Albanian language. The tagger is developed based on deep learning and achieved a high F1 score of 97%, making it a reliable tool for extracting POS-related statistics from the corpus. The tagset used in the corpus contains 58

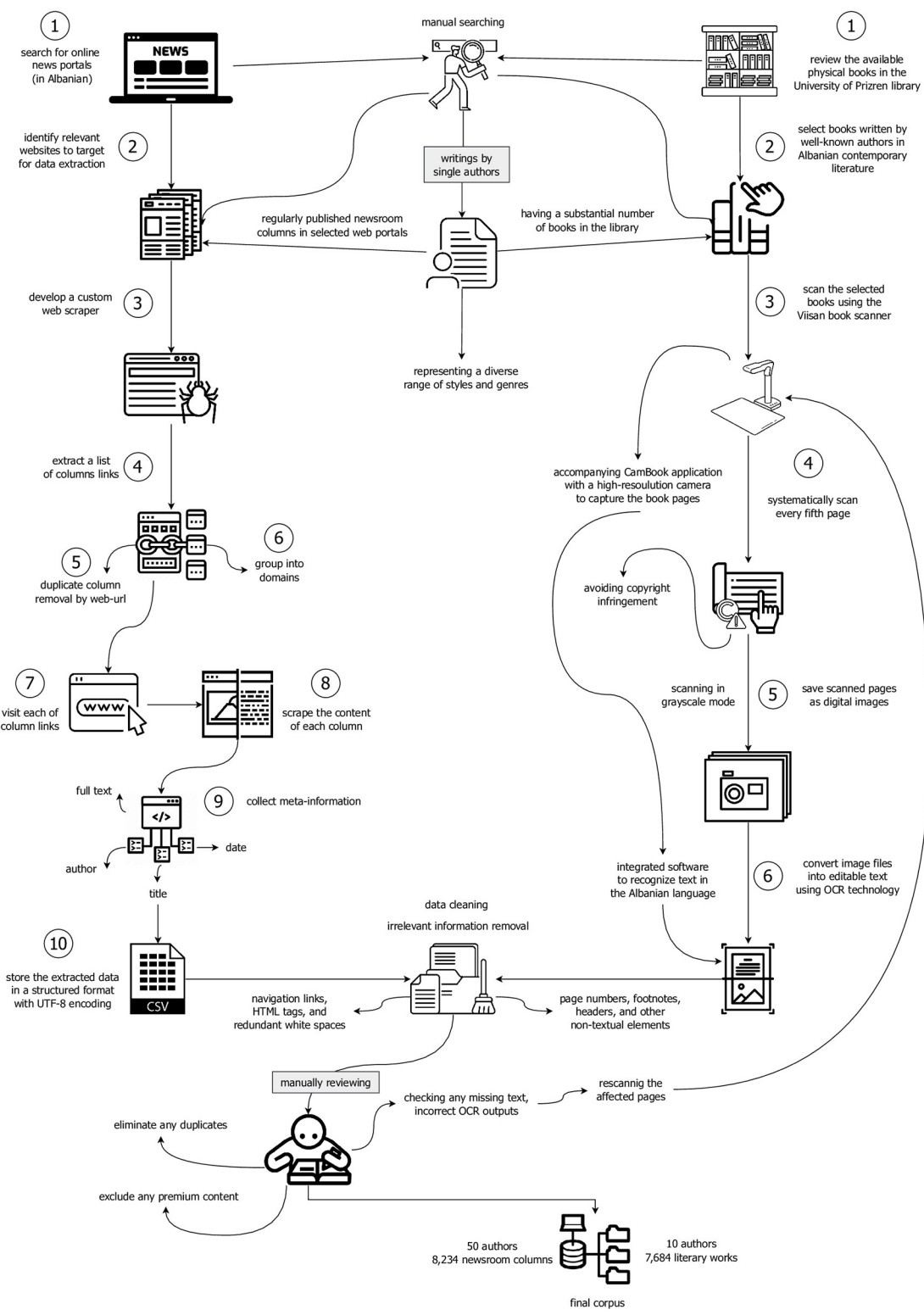

**Fig 1. Process of building the A3C-e corpus.**

**Table 2. The corpus size and authorship statistics of the A3C-e corpus.**

| Level | Features | literary works | newsroom columns |
|---|---|---|---|
| **Corpus size** | total number of samples | 7,684 | 8,234 |
| | average number of words (per sample) | 624.3 | 924.2 |
| | average number of unique words (per sample) | 305.3 | 413.8 |
| | average word length (in characters) | 5.5 | 5.9 |
| | average sentence length (in words) | 13.1 | 23.1 |
| | average unique words per sentence | 0.46 | 0.55 |
| | average number of sentences | 48.1 | 40.4 |
| | average syllables per word | 1.8 | 2.1 |
| | max length word | 63 | 55 |
| **Authorship statistics** | total number of authors | 10 | 50 |
| | average number of samples per author | 768.4 | 164.7 |
| | average number of tokens per author (per sample) | 62.43 | 18.5 |
| | average number of unique words per author (per sample) | 30.5 | 8.3 |
| | minimum total number of words per author | 185,969 | 71,494 |
| | maximum total number of words per author | 1,491,101 | 348,764 |
| | average number of words per author | 479,720 | 152,311 |
| | minimum total number of unique words per author | 14,914 | 9,741 |
| | maximum total number of unique words per author | 62,525 | 34,545 |
| | average number of unique words per author | 32,492.9 | 16,829.7 |
| **POS-related information** | lexical density (proportion of content words) | 47.84 | 49.1 |
| | average number of function words (token-based normalization) | 47.04 | 46.9 |
| | average number of verbs (token-based normalization) | 17.3 | 14.8 |
| | average number of nouns (token-based normalization) | 20.96 | 23.9 |
| | average number of adjectives (token-based normalization) | 4.51 | 6.6 |
| | average number of adverbs (token-based normalization) | 5.01 | 4.1 |
| | average number of pronouns (token-based normalization) | 9.13 | 8.3 |

POS tags, including abbreviations, adjectives, adverbs, articles, conjunctions, interjections, nouns, cardinal and decimal numbers, prefixes and suffixes, pronouns, prepositions, particles, verbs, special characters, and punctuations. The tags were assigned to each token in the corpus to provide information on the grammatical category and function of words in context.

The proposed A3C-e corpus represents a rich and extensive data collection for research and analysis. With 12,406,976 words and a vocabulary size of 4,631,714.9 unique tokens, it is a comprehensive collection of 7,684 literary works written by ten authors and 8,234 newsroom columns by 50 different authors. The corpus is a valuable resource for those researching AA in the Albanian language.

Fig 2 presents a breakdown of samples by author, showing the number of samples collected from each author and how they are distributed. This information provides a clearer picture of the dataset's composition and helps understand the data's variability.

Overall, creating this novel corpus represents an essential step toward developing automatic AA models for the Albanian language and opens up new opportunities for research in this area.

## 4 Experiments

The experiments conducted in this section aimed to investigate the effectiveness of various methods in Albanian AA. We designed a set of experiments to address the research questions

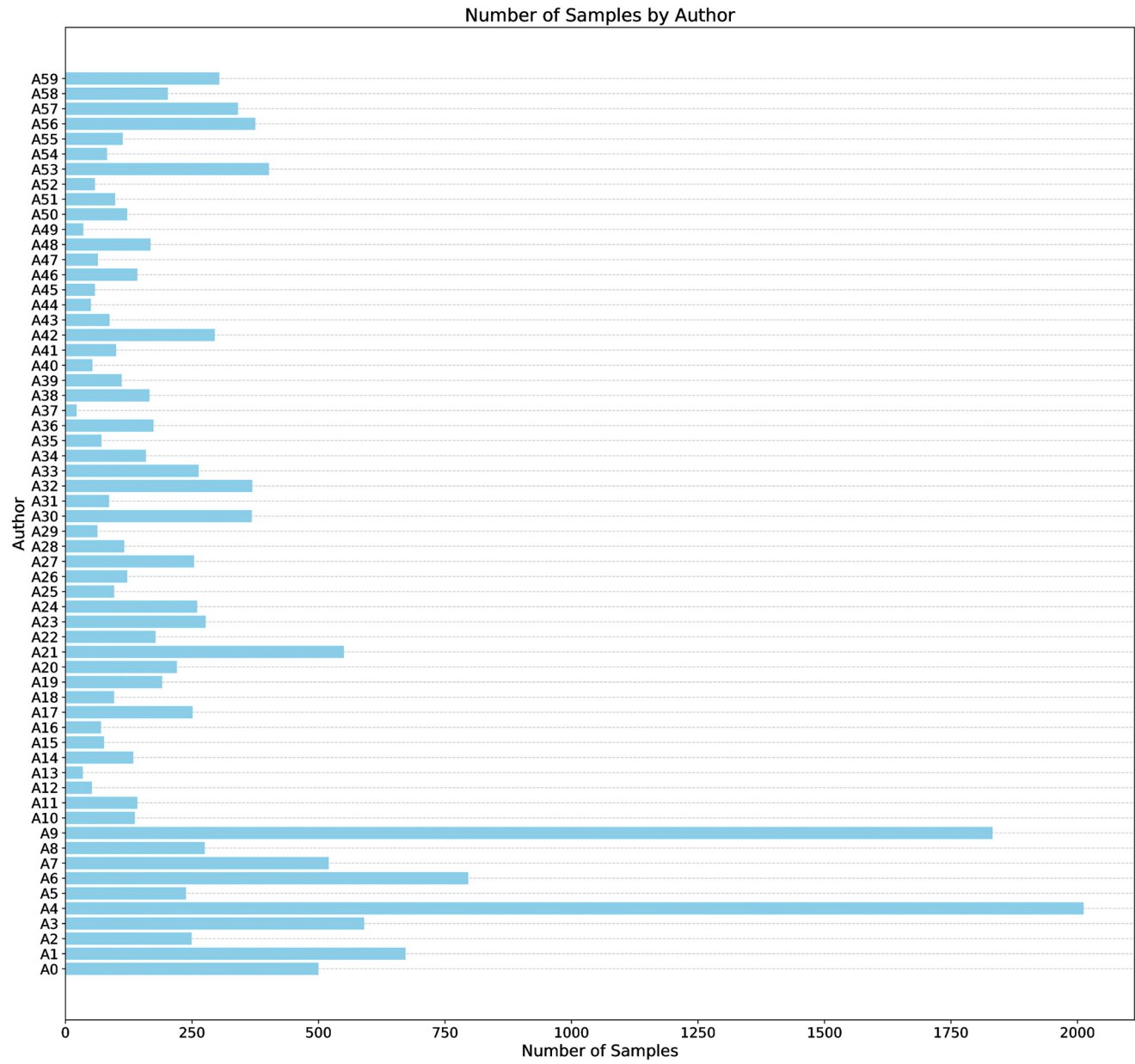

**Fig 2. Distribution of samples by author.**

formulated in Section 1. Specifically, we explored data preprocessing techniques, text representation models, feature extraction and selection methods, and multiclass classification strategies.

## 4.1 Data preprocessing

The data used for our experiments consisted of a corpus of newsroom columns and literary works written in Albanian.

The newsroom column corpus consists of articles from various Albanian news sources, while the literary corpus includes texts from published books. The dataset covers a wide range

of topics and genres, providing a diverse set of samples for authorship attribution analysis. The corpus includes a total of 15,918 samples, with 8,234 samples from the newsroom columns and 7,684 samples from the literary works. The total word count for the newsroom columns is 7,609,771, while the total word count for the literary works is 4,797,205. The average number of words per author is 206,783, with the detailed breakdown of samples per author presented in Fig 2.

**4.1.1 Data collection and compliance.** This study's data collection and analysis followed the terms and conditions of the data sources. The data was collected from publicly accessible newsroom columns and carefully selected literary works. Premium content was excluded to ensure compliance with usage restrictions. The literary works were carefully scanned to prevent copyright infringement, and a systematic method was employed to ensure compliance. All data collection procedures were conducted with appropriate measures to maintain confidentiality and proper use. The collected data was analyzed utilizing machine learning methods with the specific objective of identifying patterns in the attribution of authorship in the Albanian language. The extended Albanian Authorship Attribution Corpus is readily available to the public and may be accessed on Zenodo at 10.5281/zenodo.12699563.

The collected corpus was subjected to several preprocessing steps to ensure the quality and integrity of the data. We first performed several cleaning steps to prepare the data for analysis, as described in Section 3. Specifically, we removed irrelevant information, eliminated duplicates and null values, and corrected the errors resulting from the OCR process. Additionally, the data was processed to retain as much information and variations about the authors' handwriting styles as possible by keeping the original appearance of lowercase letters, punctuation, and other special characters.

## 4.2 Text representation models

This subsection presents the term frequency-inverse document frequency (TF-IDF) text representation model we used in our experiments. The raw text data was transformed into numerical representations using this model, which is then employed for feature extraction and classification. The TF-IDF model [4, 9] is a popular method that represents each document as a vector of term frequencies weighted by inverse document frequency.

## 4.3 Feature extraction and selection

One of the critical challenges in AA is extracting relevant features from preprocessed text data that can help distinguish between different authors.

While performing research in a low-resource language like Albanian, identifying the appropriate features for AA might be challenging. This is mainly due to the limited availability of NLP tools and resources. However, carefully, we identify relevant features that can be extracted despite the limited resources available. A combination of lexical, morphological, and structural features has been used to capture different aspects of an author's writing style. Without specific NLP tools and resources for Albanian, selecting these feature categories is a pragmatic choice that can yield valuable insights into the language's structure and usage patterns. We extract features at multiple levels—character, word, sentence, and document—to provide a comprehensive and robust feature set. The complete set of extracted features is presented in Table 3 and was carefully selected based on their relevance to the AA task.

Lexical features refer to the author's vocabulary and word usage patterns. Frequency analysis of certain words or term types can help reveal patterns in how authors use language [11, 12]. This study extracted lexical features at the character- and word levels. To extract lexical features, researchers commonly use character n-grams, where they identify and analyze groups

**Table 3. Proposed feature sets and description.**

| Category / Level | Total nr-F | Description |
|---|---|---|
| Lexical features / Character-based | 13 | total number of characters (1), lowercase letters (1), uppercase letters (1), vowels (1), consonants (1), digits (1), white spaces (1), special characters (1), punctuations (1), elongation characters (1), non-alphabetic characters (1), non-punctuation characters (1), non space characters (1) |
| | 14 | percentage of lowercase letters (1), lowercase letters-digraphs (1), uppercase letters (1), uppercase letters-digraphs (1), vowels (1), consonants (1), digits (1), white spaces (1), special characters (1), punctuations (1), elongation characters (1), non-alphabetic characters (1), non-punctuation characters (1), non space characters (1) |
| | 464 | character lowercase unigrams (36), most common character bigrams (100) and trigrams (100), character uppercase unigrams (36), digits (0-9) (10), most common unigrams (50), bigrams (50), quadrigrams (50), special characters (%, $, &) (23), punctuations (. , ? ! . . .) (9) |
| Lexical features / Word-based | 11 | total number of words (1), unique words (1), capitalized words (1), uppercase words (1), hapax legomena (1), hapax dislegomena (1), elongated words (1), syllables (1), complex words (1), long words (1), short words (1) |
| | 8 | percentage of unique words (1), capitalized words (1), uppercase words (1), hapax legomena (1), hapax dislegomena (1), elongated words (1), syllables (1), complex words (1) |
| | 550 | most frequent word uni-, bi-, and trigrams (3*100), elongated words (50), complex words (100), long words (100) |
| | 7 | vocabulary richness—type-token ratio (1), Yule's K (1), Honore's R (1), Simpson's D (1), Sichel's S (1), Brunet's W (1), Entropy measure (1) |
| | 5 | average number of characters (1), vowels (1), consonants (1), elongation characters (1), and syllables per word (1) |
| Morphological / POS-related | 21 | total number of function words (1), content words (1), stopwords (1), conjunctions (1), abbreviations (1), suffixes and prefixes (1), adjectives (1), adverbs (1), articles (1), interjections (1), nouns (1), proper nouns (1), cardinal numbers (1), pronouns (1), prepositions (1), particles (1), modal verbs (1), imperative verbs (1), past participle verbs (1), auxiliary verbs (1), verbs (1) |
| | 21 | percentage of function words (1), content words (1), stopwords (1), conjuctions (1), abbreviations (1), prefixes and suffixes (1), adjectives (1), adverbs (1), articles (1), interjections (1), nouns (1), proper nouns (1), cardinal numbers (1), pronouns (1), prepositions (1), particles (1), modal verbs (1), imperative verbs (1), past participle verbs (1), auxiliary verbs (1), verbs (1) |
| | 342 | most frequent POS unigrams (42), bigrams (100), trigrams (100), most frequent stopwords (100) |
| | 22 | average number of content words (1), function words (1), stopwords (1), conjunctions (1), abbreviations (1), suffixes and prefixes (1), unique pos types (1), adjectives (1), adverbs (1), articles (1), interjections (1), nouns (1), proper nouns (1), cardinal numbers (1), pronouns (1), prepositions (1), particles (1), modal verbs (1), imperative verbs (1), past participle verbs (1), auxiliary verbs (1), and verbs per sentence (1) |
| | 2 | lexical density (1), ratio of unique words and unique pos types (1) |
| Morphological / Inflectional forms | 7 | total number of feminine gender pos types (1), masculine gender pos types (1), singular words (1), plural words (1), 1st person pos types (1), 2nd person pos types (1), 3rd person pos types (1) |
| | 7 | average number of feminine gender pos types (1), masculine gender pos types (1), singular words (1), plural words (1), 1st person pos types (1), 2nd person pos types (1), and 3rd person pos types per sentence (1) |
| Structural | 4 | total number of lines (1), total number of blank lines (1), maximum length of words (1), minimum length of words (1) |
| Structural / Sentence-based | 5 | total number of sentences (1), sentences beginning with upper case (1), sentences beginning with lower case (1), short sentences (1), long sentences (1) |
| | 21 | average number of words (1), characters (1), white spaces (1), vowels (1), consonants (1), elongation characters (1), digits (1), special characters (1), punctuations (1), non-alphabetic characters (1), non-punctuation characters (1), non-space characters (1), unique words (1), capitalized words (1), uppercase words (1), hapax legomena (1), hapax dislegomena (1), elongated words (1), syllables (1), long words (1), and short words per sentence (1) |
| Structural / Document-based | 2 | Gunning fog readability index (1), Flesch-Kincaid readability test (1) |
| | Total: 1,526 | |

of characters to determine patterns of language use. Word n-grams, on the other hand, are groups of words that are analyzed to determine the relationship between different words in the text. Other lexical features extracted include word length, word frequency, and diversity of terms used in the text.

Moving on to morphological features, they can help identify patterns in word morphology [7], which can be particularly useful in languages like Albanian, where inflection plays an

essential role in expression meaning. This study used POS-related information [18] and inflectional word forms to extract morphological features. POS information refers to the syntactic category of a word based on its grammatical function in a sentence, such as a noun, verb, adjective, etc. Inflectional word forms involve identifying variations of words based on their person, gender, number, and other related factors. These features can capture language usage patterns characteristic of different categories of text.

Finally, this subsection discusses the extraction of structural features, which refer to the text's organization and arrangement of words and sentences. Structural features can capture the text's overall structure, such as sentence length and punctuation usage [7, 28]. These features can be significant for identifying patterns in how different authors organize and structure their writing. This study extracted structural features at the sentence- and document levels, such as sentence and document length. Additionally, we calculated the readability index of the documents. These features provide insight into different text categories' organization and structure characteristics.

We employed a combination of imperative and reflective programming paradigms to obtain a complete set of extracted features. By utilizing these two paradigms' functionality, we could identify all the user-defined functions present in the code. This process involved obtaining a symbol table of the global variables and using reflective techniques to retrieve information about the user-defined functions.

The extracted features are then used to train an ML-based model for the classification task. In addition, we present the feature selection part in the next section, which involves selecting the most relevant features that could capture important information related to AA.

## 4.4 Authorship detection methodologies

This subsection presents the authorship detection methodologies used in our experiments to perform authorship attribution. We compared two multiclass classification strategies commonly used in multiclass classification problems, specifically one-vs-one and one-vs-rest.

**4.4.1 Multiclass classification strategies.** In machine learning and classification tasks, multiclass refers to a classification problem with more than two possible classes or categories that an input can be classified into based on its features. Multiclass classification involves classifying data into more than two classes, where each sample can only be assigned to a single class label.

Two popular multiclass classification [64] strategies are used to handle multiclass problems in ML with binary classifiers: one-vs-one (OvO) and one-vs-rest (OvR). A diagram illustrating the ML-based algorithms grouped by strategy is shown in Fig 3.

OvO involves training binary classifiers for each pair of authors and combining their predictions using a voting scheme. While, OvR consists of training a binary classifier for each author and predicting the author with the highest confidence score.

OvO and OvR have their strengths and weaknesses, and the choice of algorithm will depend on the specifics of the dataset, the classifier used, and the problem that is solved. Table 4 summarizes the pros and cons of the OvO and OvR strategies for multiclass classification.

**4.4.2 Classification methods.** Different ML classification algorithms were used in our experiments. We assessed 98 ML-based models at various levels of text analysis and chose the best model in Albanian. Out of the 98, we picked 12 models that performed the best to carry out the AA task. As far as we know, these ML-based models for AA in Albanian have yet to be evaluated. Five different types of ML-based algorithms were used.

1. `tree`: Decision trees (DT) [16]

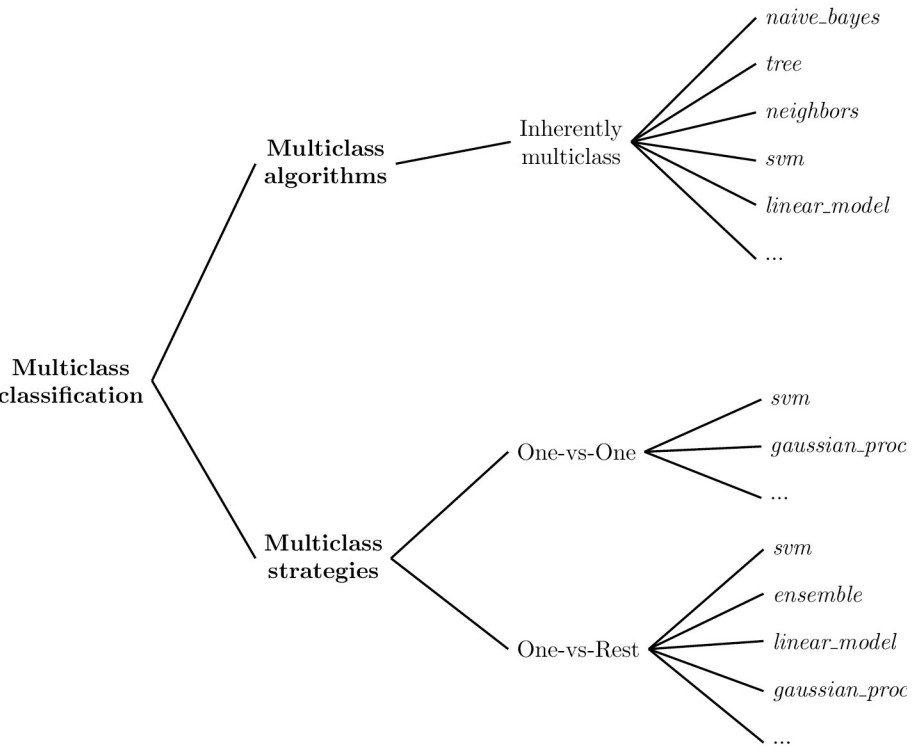

**Fig 3. Multiclass classification strategies.**

2. `svm`: Linear support vector classification (SVC) [32]

3. `ensemble`: Extreme gradient boosting (XGB) [27]

4. `discriminant_analysis`: Linear Discriminant Analysis (LDA) [65]

5. `linear_model`: Logistic Regression (LR) [12]

In the context of this work, the main focus is on evaluating the quality of the proposed corpus across various classification methods. To achieve this, we used default parameter values for the algorithms, with hyperparameter tuning left for another work. This approach allows us to assess the performance of each method in a standardized and consistent manner without the potential bias that may arise from custom parameter settings. It will ensure that our evaluation is focused exclusively on the performance of the methods rather than on optimizing their parameters.

**Table 4. Pros and cons of one-vs-one and one-vs-rest strategies for multiclass classification.**

| Multiclass strategy | Pros | Cons |
|---|---|---|
| One-vs-One | • Less sensitive to imbalanced class distributions | • Computationally expensive and time-consuming |
| | • Requires less memory | • More sensitive to overfitting |
| | • Produces more accurate predictions | • Less interpretable as it produces binary classification results |
| One-vs-Rest | • Computationally efficient | • More sensitive to unbalanced datasets |
| | • More robust when classes are poorly separable | • Can produce biased results if the dataset is imbalanced |
| | • Easier to interpret the results | • Requires more memory |

However, we performed cross-validation to evaluate the performance of each method. We trained our models using a 10-fold cross-validation approach and assessed their performance using the F1 score, the harmonic mean of precision and recall. The equation for calculating the F1 score is as follows:

$$F1\ score = 2 \times \frac{Precision \times Recall}{Precision + Recall}$$
$$= \frac{2 \times TruePositive}{2 \times TruePositive + FalsePositive + FalseNegative} \tag{1}$$

Precision is the ratio of correctly predicted positive observations to the total predicted positives, and recall is the ratio of correctly predicted positive observations to all observations in the actual class.

Since the classes are imbalanced, with some classes having more samples than others, we used the F1 score weighted averaging, with the weights being proportional to the number of samples in each class. The weighted F1 score is an extension of the F1 score that takes into account the support (the number of instances) for each class. It is particularly useful in multi-class classification problems where the class distribution is imbalanced. The weighted F1 score is computed with the following equation:

$$F1\ score\ weighted = \sum_{i=1}^{n} \frac{S_i}{S} \times F1\ score_i \tag{2}$$

where $n$ is the number of classes; $S_i$ is the support of class $i$ (the number of instances for class $i$); $S$ is the total support across all classes (the sum of all $S_i$); and $F1_i$ is the F1 score of class $i$.

A deep learning CNN model [19, 33] was also employed for AA in Albanian. The CNN model consisted of convolutional layers, a max-pooling layer, and two fully connected layers utilized for classification purposes. The activation functions designated for the convolutional and dense layers were set to 'relu' and 'softmax', respectively.

We also used the fastText model for comparison purposes. FastText [19, 33] is based on a shallow NN architecture, which is more straightforward than deep NN architectures used in DL models. The architecture of fastText includes an input layer, a hidden layer, and an output layer that uses subword information to learn vector representations for words.

Additionally, pre-trained language models like XLM-RoBERTA-base [46] and BERT-multi-lingual [43, 44] are part of our experimental setup. BERT's transformer architecture and pre-trained multilingual capabilities make it suitable for capturing contextual information in Albanian texts. This configuration enhances the robustness of our study and ensures a thorough investigation of potential models.

The results of the experiments are presented and discussed in the next section.

## 5 Results

In this section, we present the experiments conducted to evaluate the practical use and, therefore, the quality of our newly introduced corpus, namely A3C-e. This rich corpus allows us to assess the effectiveness of various ML-based algorithms in accurately attributing authorship in the Albanian language.

We also report the results of an extensive series of over a hundred experiments conducted to investigate the effectiveness of various ML-based algorithms in automatic AA for Albanian texts. Two distinct multiclass classification strategies were employed to understand better the nuances and challenges that arise when attempting to attribute authorship accurately. These

strategies were applied to diverse feature subsets and their linear combinations to identify the most promising techniques and configurations for this task.

The choice of ML-based algorithms for this study was motivated by their performance in similar tasks, their adaptability to the specific characteristics of the Albanian language, and their suitability for multiclass classification problems. By exploring various algorithms, we aimed to uncover patterns, strengths, and weaknesses that would provide valuable insights into the AA problem. This comprehensive approach allowed us better to understand the impact of various features and their interactions, ultimately leading to a more informed selection of optimal techniques for this domain.

In addition to the results of the individual experiments, we will present learning curves and class prediction errors for each algorithm and classification strategy, offering a detailed view of their performance across different training and validation stages. This in-depth analysis will enable us to assess the robustness and generalizability of our models, as well as to identify potential areas for improvement. By evaluating these aspects, we seek to contribute to the ongoing development and refinement of AA techniques, particularly for the under-researched Albanian language.

## 5.1 Experimental results

This subsection presents the classification results obtained from our extensive experimentation on the new A3C-e corpus. This comprehensive corpus enabled us to test various aspects of AA, including different types of texts, such as literary works and newsroom columns. Moreover, we considered a combined setting encompassing both types of texts to evaluate our proposed methods' robustness further.

Our experiments encompassed five ML-based algorithms, utilizing two distinct multiclass training strategies. We investigated the impact of nine different feature sets on the classification performance to ensure a broad exploration of potential feature combinations. In addition to traditional ML approaches, we experimented with two DL classification methodologies, aiming to leverage the power of neural networks for the AA task.

The A3C-e corpus was employed in three separate settings: literary works, newsroom columns, and a combination. By evaluating the performance of our algorithms in these distinct contexts, we aimed to assess their adaptability and effectiveness across a diverse range of texts. Furthermore, we conducted experiments to identify the minimum feature set required for successful AA. It will help refine the process and reduce the computational complexity of the models without sacrificing accuracy.

In the following paragraphs, we will delve into the details of our experimental results, comparing the performance of various ML-based algorithms, feature sets, and training strategies. We will also discuss the effectiveness of DL methodologies in the context of AA and explore the implications of our findings for identifying the most efficient feature sets. This comprehensive analysis aims to provide valuable insights into the state-of-the-art methods for AA in Albanian texts.

Table 5 presents the results of the first series of experiments. In it, we compare the F1 scores of different classification algorithms and various sets of features. Regarding the input data in this first experiment, we use the subset of our original corpus, namely the literary works.

When the standard TD-IF schema was used for feature extraction, the best results were archived with SVC. The highest score of 0.99 is archived with the following parameters: 1) a multiclass training strategy is one-vs-rest, 2) the TF-IDF features without stop-words removal, and 3) the algorithm is SVC. It is well known that stop word removal helps in increasing the accuracy of text classification models. However, removing stop words may not improve

**Table 5. Classification results of ML-based algorithms for literary works—A comparison of accuracy.** Best results for each category of experiments are given in bold.

| Set of features | One-vs-One | | | | | One-vs-Rest | | | | |
|---|---|---|---|---|---|---|---|---|---|---|
| | DT | SVC | XGB | LDA | LR | DT | SVC | XGB | LDA | LR |
| TF-IDF (No-SW) | 0.869 | **0.981** | 0.936 | 0.893 | **0.981** | 0.753 | **0.990** | 0.958 | 0.982 | 0.984 |
| TF-IDF (With-SW) | 0.839 | 0.969 | 0.914 | 0.855 | 0.958 | 0.753 | 0.969 | 0.939 | 0.955 | 0.955 |
| LF | 0.876 | 0.897 | 0.951 | 0.865 | 0.940 | 0.793 | 0.867 | 0.972 | 0.950 | 0.959 |
| MF | 0.752 | 0.905 | 0.907 | 0.898 | 0.889 | 0.673 | 0.900 | 0.931 | 0.906 | 0.907 |
| SF | 0.720 | 0.651 | 0.816 | 0.729 | 0.784 | 0.674 | 0.628 | 0.834 | 0.601 | 0.713 |
| LF + MF | 0.882 | 0.913 | 0.959 | 0.853 | 0.952 | 0.806 | 0.923 | 0.978 | 0.969 | 0.973 |
| LF + SF | 0.878 | 0.896 | 0.952 | 0.869 | 0.944 | 0.809 | 0.899 | 0.974 | 0.950 | 0.960 |
| MF + SF | 0.810 | 0.910 | 0.924 | 0.911 | 0.902 | 0.717 | 0.915 | 0.947 | 0.913 | 0.926 |
| LF + MF + SF | 0.888 | 0.923 | **0.970** | 0.854 | 0.967 | 0.811 | 0.919 | **0.982** | 0.971 | 0.980 |

**TF-IDF** = Term Frequency-Inverse Document Frequency; **No-SW** = No Stop-Words; **With-SW** = With Stop-Words; **LF** = Lexical Features; **MF** = Morphological Features; **SF** = Strutural Features; **DT** = Decision Tree; **SVC** = Linear Support Vector Classifier; **XGB** = XGBoost; **LDA** = Linear Discriminant Analysis; **LR** = Logistic Regression.

classification results in some cases. It could be due to several reasons, such as the stop words significantly impacting the meaning or context of the text, or the classifier can handle them effectively. Additionally, the frequency of stop words may need to be higher in the dataset to impact the TF-IDF scores significantly. We strongly believe that the stop words in the case of AA positively impact distinguishing the authors, which is also supported by results presented in Table 5.

On the contrary, experimental outcomes employing manually engineered attributes yield the most favorable results when the complete array of features is implemented. In this instance, the XGBoost algorithm utilizing the OvR multiclass training methodology, combined with a combination of lexical, morphological, and structural attributes, achieves the highest accuracy rate of 0.982.

Analogous to the results delineated in the preceding table, Table 6 exhibits the same experimental configuration, albeit with the AA task applied to the newsroom columns subset within

**Table 6. Classification results of ML-based algorithms for newsroom columns—A comparison of accuracy.** The best results for each category of experiments are given in bold.

| Set of features | One-vs-One | | | | | One-vs-Rest | | | | |
|---|---|---|---|---|---|---|---|---|---|---|
| | DT | SVC | XGB | LDA | LR | DT | SVC | XGB | LDA | LR |
| TF-IDF (No-SW) | 0.801 | **0.917** | 0.832 | 0.572 | **0.917** | 0.500 | **0.944** | 0.883 | 0.907 | 0.929 |
| TF-IDF (With-SW) | 0.741 | 0.886 | 0.754 | 0.542 | 0.893 | 0.455 | 0.900 | 0.818 | 0.872 | 0.895 |
| LF | 0.731 | 0.487 | 0.789 | 0.573 | 0.789 | 0.491 | 0.434 | 0.853 | 0.831 | 0.876 |
| MF | 0.712 | 0.666 | 0.769 | 0.580 | 0.761 | 0.404 | 0.689 | 0.836 | 0.851 | 0.844 |
| SF | 0.560 | 0.503 | 0.607 | 0.643 | 0.646 | 0.381 | 0.286 | 0.635 | 0.542 | 0.485 |
| LF + MF | 0.802 | 0.571 | **0.845** | 0.605 | 0.828 | 0.521 | 0.601 | **0.905** | 0.890 | 0.901 |
| LF + SF | 0.753 | 0.584 | 0.803 | 0.591 | 0.822 | 0.511 | 0.343 | 0.872 | 0.854 | 0.871 |
| MF + SF | 0.743 | 0.754 | 0.793 | 0.566 | 0.782 | 0.451 | 0.720 | 0.865 | 0.873 | 0.873 |
| LF + MF + SF | 0.808 | 0.651 | 0.841 | 0.733 | 0.799 | 0.558 | 0.625 | 0.899 | 0.874 | 0.871 |

**TF-IDF** = Term Frequency-Inverse Document Frequency; **No-SW** = No Stop-Words; **With-SW** = With Stop-Words; **LF** = Lexical Features; **MF** = Morphological Features; **SF** = Strutural Features; **DT** = Decision Tree; **SVC** = Linear Support Vector Classifier; **XGB** = XGBoost; **LDA** = Linear Discriminant Analysis; **LR** = Logistic Regression.

**Table 7. Classification results of ML-based algorithms for literary works and newsroom columns combined—A comparison of accuracy.** The best results for each category of experiments are given in bold.

| Set of features | One-vs-One | | | | | One-vs-Rest | | | | |
|---|---|---|---|---|---|---|---|---|---|---|
| | DT | SVC | XGB | LDA | LR | DT | SVC | XGB | LDA | LR |
| TF-IDF (No-SW) | 0.818 | 0.938 | 0.869 | 0.674 | **0.940** | 0.593 | **0.953** | 0.921 | 0.931 | 0.947 |
| TF-IDF (With-SW) | 0.786 | 0.903 | 0.813 | 0.588 | 0.894 | 0.524 | 0.914 | 0.844 | 0.883 | 0.915 |
| LF | 0.804 | 0.661 | 0.872 | 0.622 | 0.881 | 0.621 | 0.592 | 0.906 | 0.871 | 0.912 |
| MF | 0.731 | 0.764 | 0.821 | 0.704 | 0.831 | 0.501 | 0.752 | 0.876 | 0.841 | 0.881 |
| SF | 0.637 | 0.532 | 0.703 | 0.677 | 0.696 | 0.493 | 0.333 | 0.713 | 0.455 | 0.577 |
| LF + MF | 0.833 | 0.711 | 0.886 | 0.708 | 0.884 | 0.641 | 0.742 | 0.930 | 0.913 | 0.926 |
| LF + SF | 0.802 | 0.723 | 0.872 | 0.644 | 0.884 | 0.631 | 0.681 | 0.915 | 0.882 | 0.913 |
| MF + SF | 0.762 | 0.824 | 0.864 | 0.737 | 0.844 | 0.540 | 0.814 | 0.903 | 0.873 | 0.911 |
| LF + MF + SF | 0.850 | 0.753 | **0.903** | 0.716 | 0.894 | 0.644 | 0.724 | **0.943** | 0.922 | 0.941 |

**TF-IDF** = Term Frequency-Inverse Document Frequency; **No-SW** = No Stop-Words; **With-SW** = With Stop-Words; **LF** = Lexical Features; **MF** = Morphological Features; **SF** = Strutural Features; **DT** = Decision Tree; **SVC** = Linear Support Vector Classifier; **XGB** = XGBoost; **LDA** = Linear Discriminant Analysis; **LR** = Logistic Regression.

the original corpus. We discern comparable outcomes in this scenario, particularly concerning the TF-IDF attribute set. The most striking results are discerned in the context of hand-crafted features. Rather than employing a comprehensive set amalgamated with all extracted attributes, which amounts to over 1500 attributes, solely the subset of lexical and morphological features trained with XGBoost performed optimally in terms of accuracy.

We construe this discovery based on our domain-specific expertise. Given that newsroom columns predominantly focus on facts and information, the AA model would derive the most advantage from the utilized vocabulary, as opposed to writing style or analogous signals more closely related to literary work.

In the last experiment of this series, the entire corpus is analyzed, with results presented in Table 7. These findings are consistent with the outcomes delineated in the preceding two tables.

In the following experiments (Table 8), we evaluated our recently developed corpus on the training of DL-based models. For this purpose, we incorporated various DL training models, fastText, CNN, and Transformers-based pre-trained models like BERT-multilingual and XLM-RoBERTa-base, to demonstrate their efficacy in AA.

The F1 scores achieved using fastText, CNN, and BERT models were observed to be marginally lower than those attained with TF-IDF and manually engineered features when

**Table 8. Classification results of DL-based models in three separate settings (literary works, newsroom columns, and a combination).**

| Corpus | fastText | | CNN | | Transformers | | | |
|---|---|---|---|---|---|---|---|---|
| | | | | | BERT-multilingual | | XLM-RoBERTa | |
| | With-SW | No-SW | With-SW | No-SW | With-SW | No-SW | With-SW | No-SW |
| **literary works** | 0.977 | 0.947 | 0.941 | 0.925 | 0.952 | 0.966 | 0.962 | 0.976 |
| **columns** | 0.904 | 0.861 | 0.633 | 0.626 | 0.814 | 0.805 | 0.851 | 0.831 |
| **literary and columns** | 0.934 | 0.904 | 0.742 | 0.721 | 0.884 | 0.888 | 0.881 | 0.908 |

**No-SW** = No Stop-Words; **With-SW** = With Stop-Words; **CNN** = Convolutional Neural Network.

classifying literary works. It indicates that DL-based models may still need to outperform traditional ML techniques in literary works classification.

In the case of newsroom columns, the DL classification models, particularly the CNN model, reported significantly lower performance metrics than the previously obtained measurements. This outcome suggests that the CNN model is not optimally suited for the classification of authorship in the newsroom columns corpus, potentially due to the data's unique nature and the texts' underlying structure.

When the entire corpus is considered, the fastText model demonstrates commendable performance, with results nearly on par with those reported in earlier experiments. This outcome highlights the potential of the fastText model in capturing and generalizing across diverse textual datasets. BERT also achieved notable accuracy improvements in specific scenarios. In contrast, the CNN model exhibited the least favorable F1 scores, which could be attributed to their architectural limitations or the need for further optimization of hyperparameters to improve their performance in the AA domain.

As the XGBoost algorithm consistently provided better results across the three separate corpus subsets and various feature combinations in our experiments, we used it as the classifier for subsequent experiments.

## 5.2 Feature analysis and selection

This subsection analyzes features used to classify authorship in Albanian written texts. Using SHAP values [66] and generating waterfall plots, we examine the importance of individual hand-crafted and model-generated features and their impact on the author classification task. Each bar in the SHAP waterfall plots represents a feature, with its length indicating its impact on the model's output for a specific example. Classification models rely on an ensemble of features and their linear combinations, with no single attribute determining the classification outcome. Figs 4 and 5 show that sentence-level structural features, morphological features represented by POS tags, as well as certain nouns and adverbs, had the most significant impact on authorship classification.

Considering the extensive array of manually crafted features extracted to enhance model accuracy, we sought to examine the influence of feature selection on the performance of the trained XGBoost model in terms of classification results in the subsequent series of experiments.

Feature importance scores generated by the XGBoost model help identify the features with the most significant authorship information, guiding the selection process. We performed feature selection analysis on the combined dataset using the XGBoost classifier with hand-crafted and TF-IDF features.

This subsection describes the iterative feature elimination process [67] and its impact on model performance. The x-axis represents the number of features, while the y-axis shows the model's cross-validated score, illustrating how performance evolves with different feature counts. The analysis identifies two key points: the peak performance number of features and a threshold number of features where performance drops by 0.02 from the peak, marked by a dashed line. This threshold indicates the balance between feature inclusion and model reliability.

The analysis in Figs 6 and 7 helps identify optimal feature configurations for the dataset, systematically evaluating how different feature subsets impact the model's performance. This observation suggests that employing a reduced set of features may not adversely impact the model performance, and a more concise feature set may still yield reliable classification results.

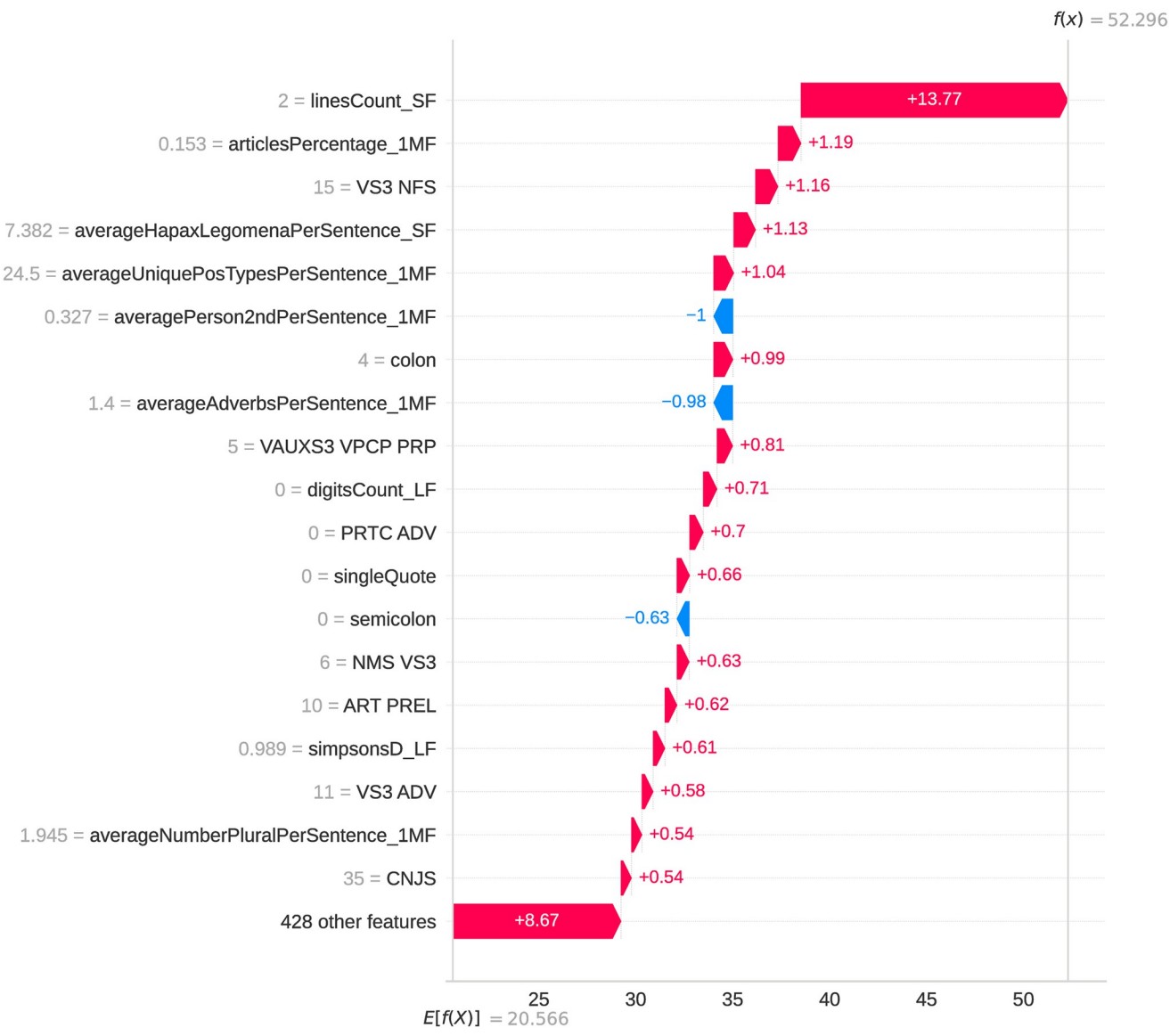

**Fig 4. SHAP waterfall plot for a specific example in the combined dataset using XGBoost algorithm (hand-crafted features).**

## 5.3 Learning curves

In this subsection, we present the learning curves to show the correlation between the training score and the cross-validated test score for the XGBoost estimator using the OvR multiclass classification strategy with different training sample sizes.

Since the training score in Figs 8 and 9 is greater than the validation score, and the difference between them gradually decreases over time, it suggests that the model is slowly beginning to generalize better to the unseen data. This improvement may be attributed to an increase in the training data size or the inclusion of more diverse examples, which allows the model to learn more representative patterns of the underlying data distribution.

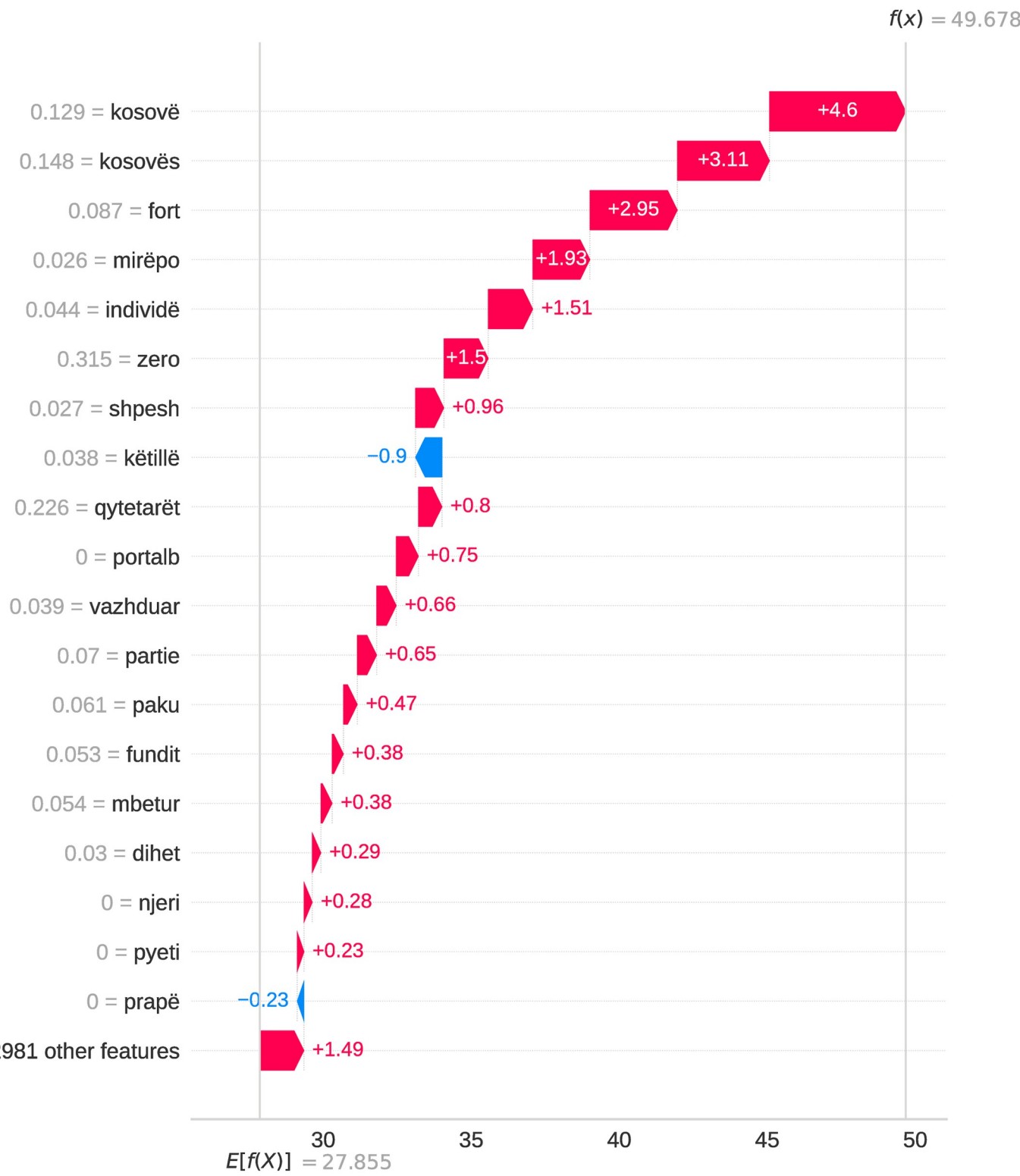

**Fig 5. SHAP waterfall plot for a specific example in the combined dataset using XGBoost algorithm (TF-IDF features).**

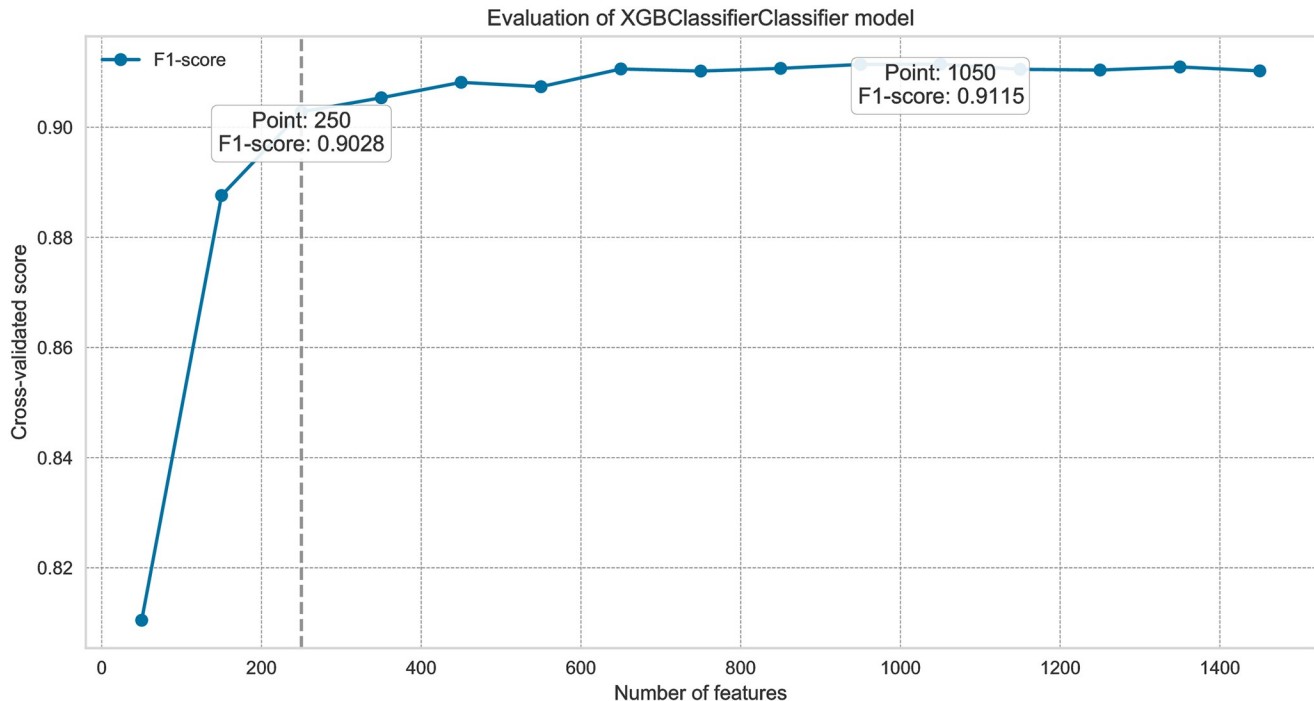

**Fig 6. The relationship between feature set selection and the model's performance across literary and column contexts (hand-crafted features).**

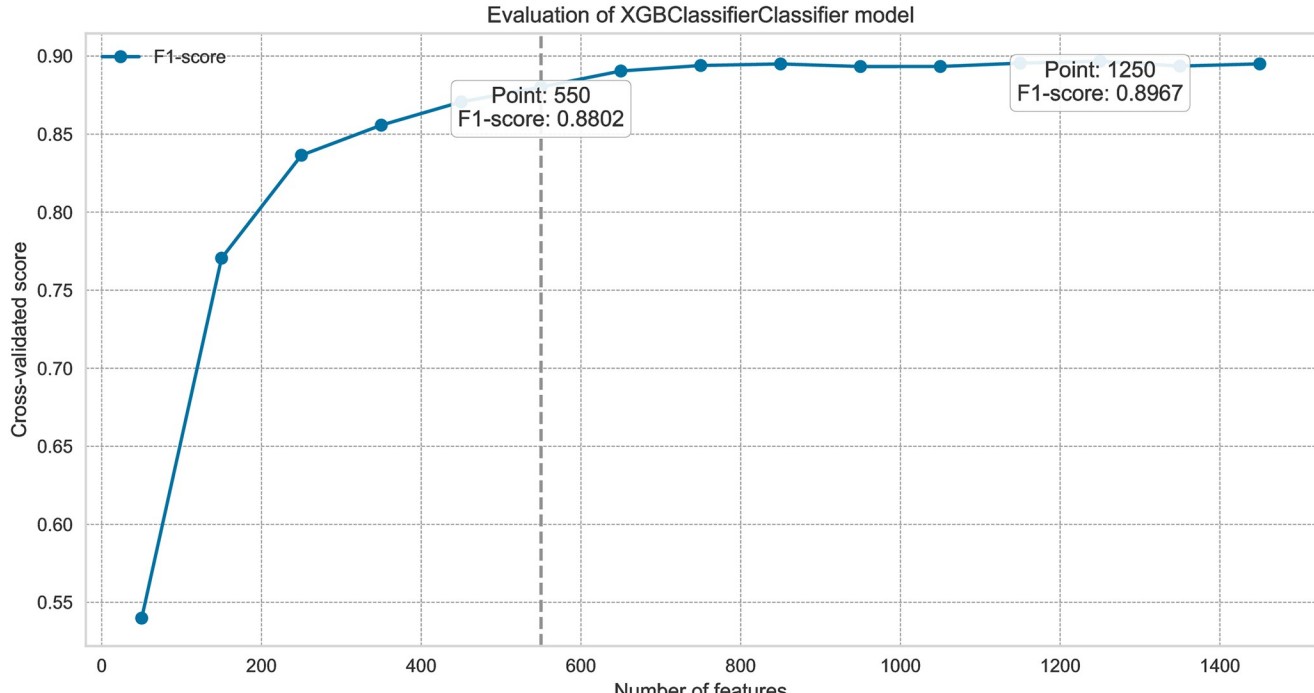

**Fig 7. The relationship between feature set selection and the model's performance across literary and column contexts (TF-IDF features).**

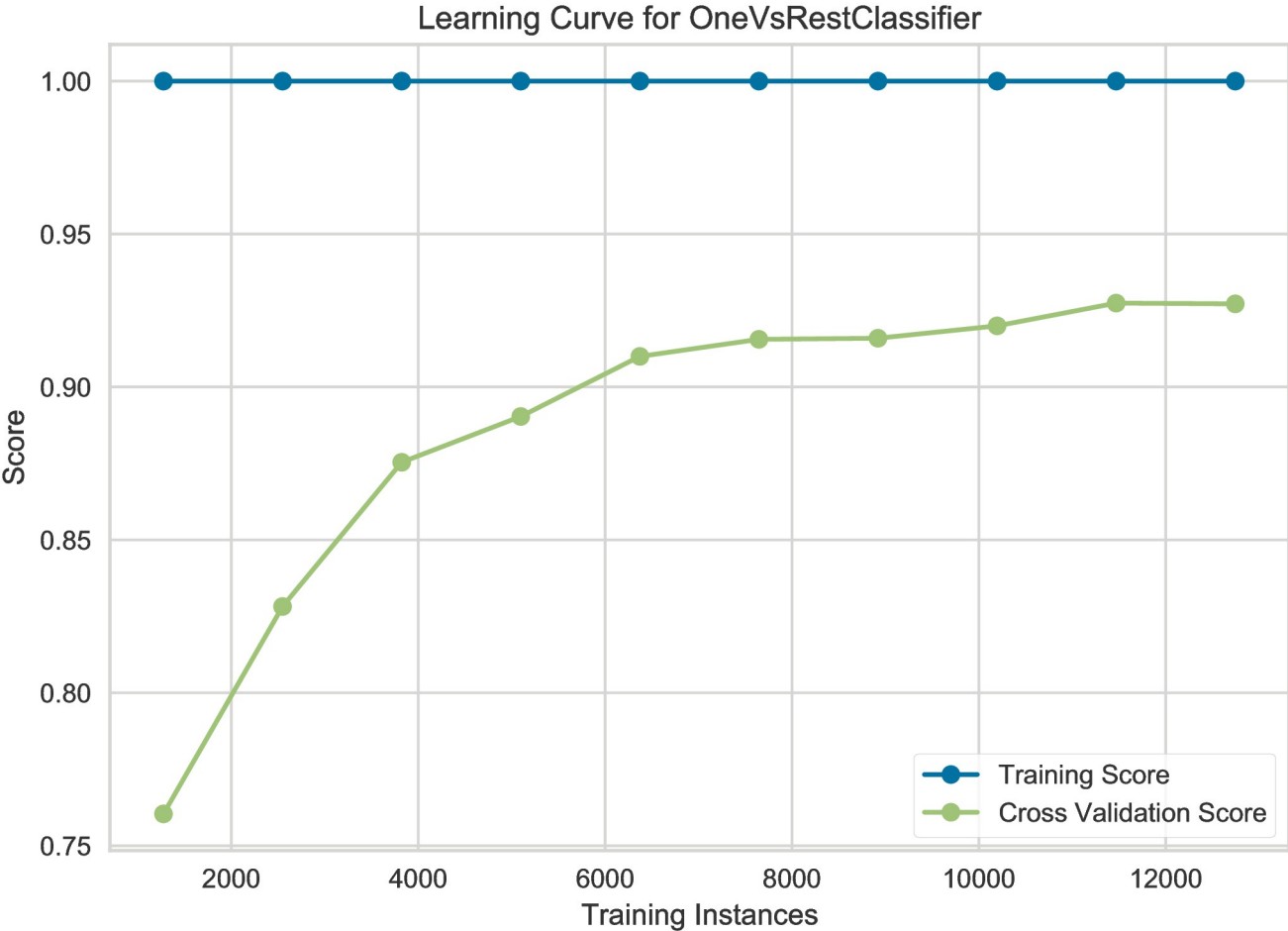

**Fig 8. Learning curve for the XGBoost estimator of the OvR multiclass classification strategy (hand-crafted features).**

### 5.4 Class prediction error analysis

In this subsection, we comprehensively analyze class prediction error to investigate ML-based models' efficacy in accurately determining a text's authorship.

This analysis is visually presented in the following figures, offering an alternative method of displaying the confusion matrix, a tabular representation that outlines the performance of a classification model by enumerating the correctly and incorrectly predicted instances for each author. Given the substantial number of classes involved, employing a standard confusion matrix would result in unwieldy matrices; hence, we opt for class prediction and continuous error plots as our mode of communication.

A comparison of error rates in Figs 10 and 11 reveals that models trained with hand-crafted features exhibit a lower average deviation compared to those trained on TF-IDF features.

Figs 12 and 13 exhibit a pattern analogous to the one described here. In Fig 13, which depicts the continuous error plot for the XGBoost classifier, a noticeably greater distribution of blue cells can be seen in comparison to Fig 12, where the same model is trained on hand-crafted features.

Based on the error rate and continuous error plots, we conclude that despite the classification models trained on TF-IDF representations displaying superior F1 scores, a more in-depth analysis of error rates reveals enhanced robustness in models trained on hand-crafted features.

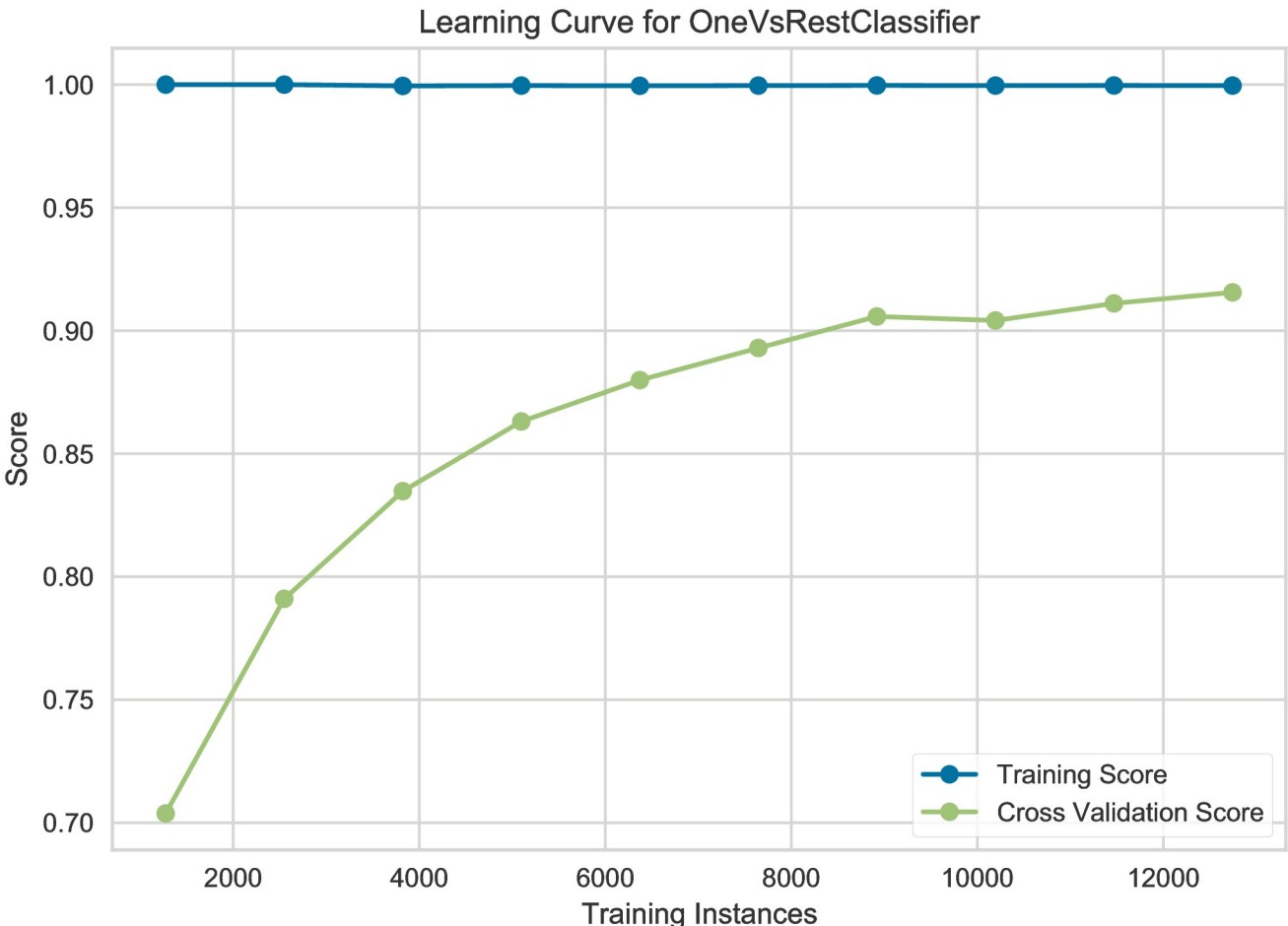

**Fig 9. Learning curve for the XGBoost estimator of the OvR multiclass classification strategy (TF-IDF features).**

## 6 Conclusions

In this paper, we investigated the problem of authorship attribution in Albanian by introducing a newly compiled corpus and conducting an in-depth analysis of ML-based methods for detecting authorship. We experimented with various hand-crafted features and applied traditional and DL-based models to determine the optimal combination of features and classification methods for automatic AA. Our results provide valuable insights, as we have carefully crafted our experiments to answer the following research questions.

**RQ** 1: *What are the most effective linguistic features for accurately identifying the authors in Albanian texts?*

We conducted several experiments utilizing manually engineered features and those crafted from a text representation model. The results indicated that the TF-IDF model outperformed the linguistic features, but to varying degrees depending on the specific algorithm used. However, generally speaking, the results of these experiments showed that, in most cases, there were no significant differences between using the two feature sets. Overall, these findings suggest that manually engineered and text representation model-crafted features can be practical in specific contexts, and the choice between them should depend on the specific needs of the task.

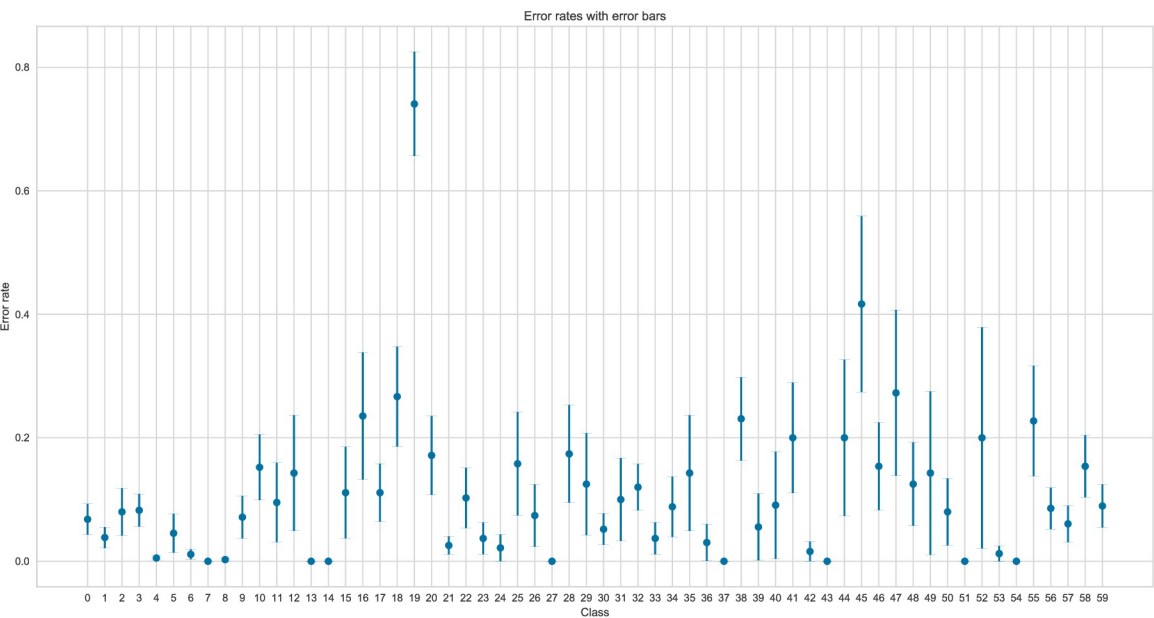

**Fig 10. Error rates for the XGBoost classifier trained with hand-crafted features.**

**RQ** *2*: *Can machine learning approaches be effectively used for authorship attribution in Albanian language texts, and if so, which algorithms perform the best?*
Five traditional ML-based algorithms (DT, SVC, XGBoost, LDA, and LR) and four DL-based classification models (fastText, CNN, BERT-multilingual, and XLM-RoBERTa) were used to evaluate their performance. In all experiments, XGBoost and LR demonstrated their superiority. While the DT algorithm generally obtained the lowest results. Further, we

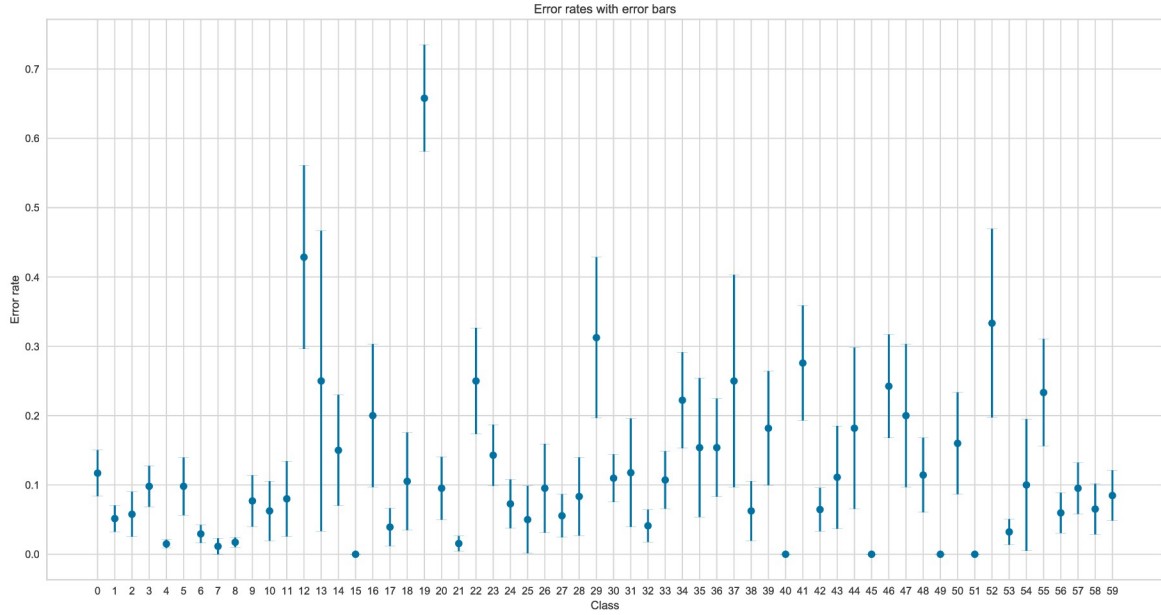

**Fig 11. Error rates for the XGBoost classifier trained with TF-IDF features.**

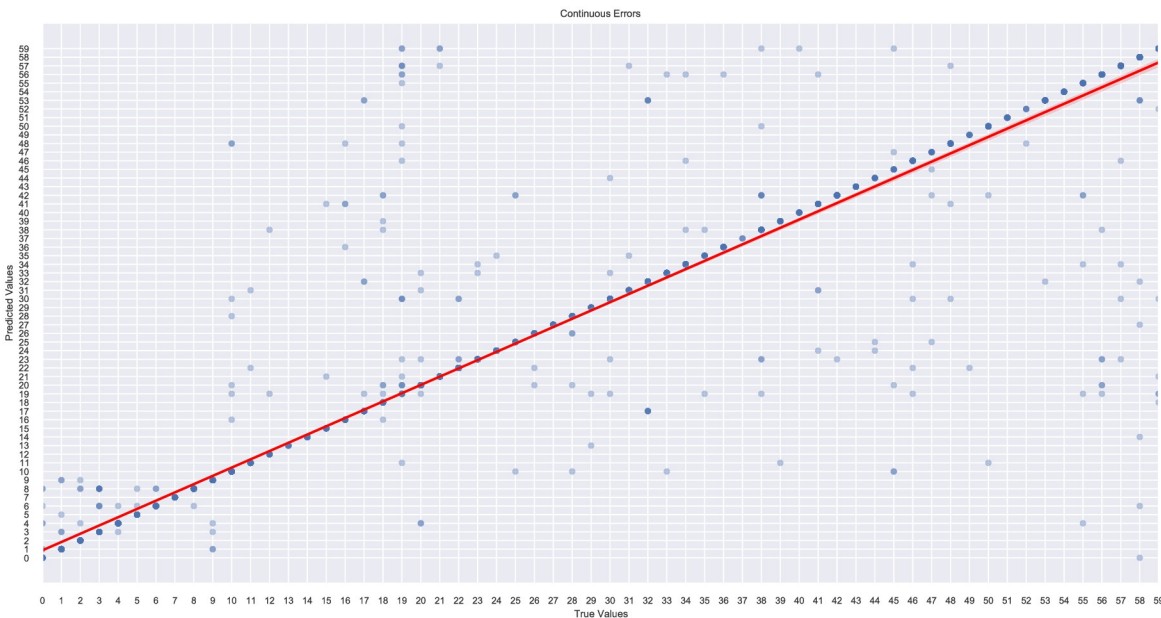

**Fig 12. The correctly and incorrectly predicted instances for each author—The performance of the XGBoost classifier (hand-crafted features).**

observed that the results obtained with DL-based models were slightly lower than those of ML-based algorithms when comparing NN to traditional classifiers. Still, the fastText and the BERT-based models demonstrated impressive performance, producing results nearly comparable to ML-based algorithms.

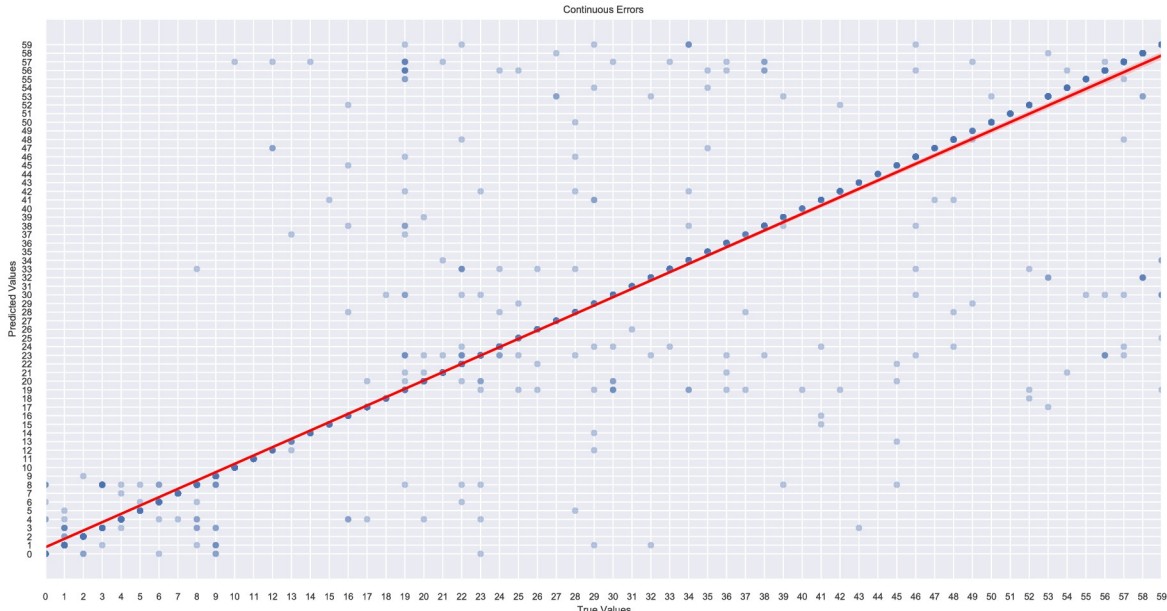

**Fig 13. The correctly and incorrectly predicted instances for each author—The performance of the XGBoost classifier (TF-IDF features).**

**RQ** *3*: *What are the effects of multiclass classification strategies on authorship attribution?*

In our study, we investigated the performance of multiclass classification in two different strategies: OvO and OvR. Five different ML-based classifiers were selected as estimators in each strategy to evaluate how each algorithm handles multiclass classification with binary classifiers. We used the same set of algorithms for each strategy to ensure consistency in the evaluation. The findings generally present a reliable and steady performance across various feature subsets and their combinations in different text types. Based on the experimental results, it is evident that there was a marked inconsistency in the F1 scores of the LDA and DT classifiers, mainly when using the TF-IDF attribute set. The LDA algorithm gave significantly better results when using the OvR strategy, while DT performed better in the OvO multiclass classification strategy. In contrast, LR and XGBoost algorithms were found to be consistently effective across both multiclass classification strategies, with the OvR strategy producing the best outcomes. Furthermore, SVC exhibited better results when using the TF-IDF feature set in both multiclass classification strategies. Interestingly, while the SVC classifier using manually crafted features performed better in the OvO strategy, the algorithm prevailed using the TF-IDF feature set in the OvR strategy.

The results of this study show promising performance for AA, but there are still many areas for further research. Firstly, we aim to evaluate our approach to other authorship-related tasks, such as author profiling, and investigate its effectiveness in verifying authorship in Albanian writings. To expand the scope of the research, we will investigate authorship classification on an extended data set with more writings from various authors of different genres, including short messages from social users. To further improve the proposed approach, we plan to explore the use of ensembles of classifiers, transformers, and other powerful learning techniques to identify authors for respective writings. Lastly, we intend to explore additional features for authorship-related tasks, including stylometric, syntactic, domain-specific, and psycho-linguistic features. These features will be further investigated and integrated into the author identification models.

Our future research will lead to a better understanding of the different factors that influence authorship identification and will help improve AA models' accuracy and robustness in various contexts and domains.

## Author Contributions

**Conceptualization:** Arta Misini, Ercan Canhasi, Arbana Kadriu, Endrit Fetahi.

**Data curation:** Arta Misini.

**Formal analysis:** Arta Misini.

**Investigation:** Arta Misini.

**Methodology:** Arta Misini, Ercan Canhasi, Arbana Kadriu, Endrit Fetahi.

**Project administration:** Arta Misini, Ercan Canhasi, Arbana Kadriu.

**Resources:** Arta Misini, Ercan Canhasi, Arbana Kadriu, Endrit Fetahi.

**Software:** Arta Misini, Ercan Canhasi.

**Supervision:** Arta Misini, Ercan Canhasi, Arbana Kadriu.

**Validation:** Ercan Canhasi, Arbana Kadriu, Endrit Fetahi.

**Visualization:** Arta Misini.

**Writing – original draft:** Arta Misini, Ercan Canhasi.

**Writing – review & editing:** Arta Misini, Ercan Canhasi, Arbana Kadriu.

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
