## [Decision Letter · Decision Letter 0]

25 Jul 2024

PONE-D-24-28247Automatic Authorship Attribution in Albanian textsPLOS ONE

Dear Dr. Misini,

Thank you for submitting your manuscript to PLOS ONE. After careful consideration, we feel that it has merit but does not fully meet PLOS ONE’s publication criteria as it currently stands. Therefore, we invite you to submit a revised version of the manuscript that addresses the points raised during the review process.

We look forward to receiving your revised manuscript.

Kind regards,

Muhammad Afzaal, PhD

Academic Editor

PLOS ONE

Journal Requirements:

2. In your Methods section, please include additional information about your dataset and ensure that you have included a statement specifying whether the collection and analysis method complied with the terms and conditions for the source of the data.

3. Please note that PLOS ONE has specific guidelines on code sharing for submissions in which author-generated code underpins the findings in the manuscript. In these cases, we expect all author-generated code to be made available without restrictions upon publication of the work.

Please review our guidelines at https://journals.plos.org/plosone/s/materials-and-software-sharing#loc-sharing-code and ensure that your code is shared in a way that follows best practice and facilitates reproducibility and reuse.

4. Please note that your Data Availability Statement is currently missing the direct link to access each database. If your manuscript is accepted for publication, you will be asked to provide these details on a very short timeline. We therefore suggest that you provide this information now, though we will not hold up the peer review process if you are unable.

Reviewers' comments:

Reviewer's Responses to Questions

**Comments to the Author**

1. Is the manuscript technically sound, and do the data support the conclusions?

Reviewer #1: Yes

Reviewer #2: Yes

2. Has the statistical analysis been performed appropriately and rigorously? 

Reviewer #1: Yes

Reviewer #2: Yes

3. Have the authors made all data underlying the findings in their manuscript fully available?

Reviewer #1: Yes

Reviewer #2: Yes

4. Is the manuscript presented in an intelligible fashion and written in standard English?

Reviewer #1: Yes

Reviewer #2: Yes

5. Review Comments to the Author

Reviewer #1: This study presents a statistically robust investigation into the authorship attribution of the low-resource language Albabian through a strategic employment of multiclass classification techniques and machine learning approaches on chosen sets of linguistic features. The subsections of the article are neatly organized with an elaborated description of the analysis procedure and a clear presentation of experimental results. The insights derived from the results are also interesting. However, several points need to be addressed:

1. Clarification of Results in the Abstract Section: The authors are recommended to summarize concrete results in the abstract section, specifying the most suitable set of linguistic features for authorship attribution and the best machine learning approaches for Albanian texts, as proposed in the research questions.

2. Possible Effect of “Average Number of Words (Tokens) per Author” on ML Algorithm Performance: In terms of data size for authorship attribution tasks, 10,000 words per author is considered the “reliable minimum for an authorial set” (Burrows, 2007). As presented in Table 2, “Authorship Statistics,” the average number of tokens per author in both genres is far too limited when measured against this standard. The effect of factors such as “number of candidate authors” and “data size” has been fully acknowledged by studies such as Luyckx & Daelemans (2011). Could the performance of authorship attribution in the present study also be influenced by factors like “author set size”?

Reviewer #2: This paper introduced the extended Albanian authorship attribution corpus (A3C-e) and the most effective linguistic features and the most effective machine learning models. It took considerable efforts to build the Albanian authorship attribution corpus, consisting of the newsroom text and scanned books. Extensive experiments were also conducted to identify the most effective linguistic features and machine learning models.

However, there is still some room for improvement:

1. As an authorship attribution corpus, it’s better to include of a breakdown of samples by author. For example, how many samples are collected from one author? How are the samples distributed by authors?

2. It’s better to detail the imperative and reflective programming paradigms that were used to extract features. Are there any open-source tools used? Will the codes for feature extraction be released publicly or not?

3. For deep learning methods, BERT multilingual also support Albanian. It’s better to include BERT in the deep learning methods.

4. It’s better to define F1, that is, the equation for F1 calculation.

6. PLOS authors have the option to publish the peer review history of their article (what does this mean?). If published, this will include your full peer review and any attached files.

Reviewer #1: **Yes: **Xiao Shanshan

Reviewer #2: No

---

## [Author Response · Author response to Decision Letter 0]

10 Aug 2024

Response to Review Comments

(Manuscript number: PONE-S-24-36426)

We would like to express our sincere gratitude to you for the time and effort you have dedicated to reviewing our manuscript, titled "Automatic Authorship Attribution in Albanian texts," ID PONE-S-24-36426. We appreciate your constructive feedback, which has been invaluable in improving the quality and clarity of our work. Furthermore, we value you assigning reviewers to our work in such a short time.

We have carefully considered each of your suggestions and have made the necessary revisions to address the points raised. Our responses to your comments are detailed below.

Academic Editor

Response:

Done!

2. In your Methods section, please include additional information about your dataset and ensure that you have included a statement specifying whether the collection and analysis method complied with the terms and conditions for the source of the data.

Response:

Thank you for your valuable feedback. In response to this comment, we have revised the Experiments section to include additional details about the dataset and a compliance statement (subsection 4.1 Data preprocessing). This study's data collection and analysis adhered to the terms and conditions of the data sources.

3. Please note that PLOS ONE has specific guidelines on code sharing for submissions in which author-generated code underpins the findings in the manuscript. In these cases, we expect all author-generated code to be made available without restrictions upon publication of the work.

Please review our guidelines at https://journals.plos.org/plosone/s/materials-and-software-sharing#loc-sharing-code and ensure that your code is shared in a way that follows best practice and facilitates reproducibility and reuse.

Response:

Thank you for bringing the PLOS ONE code-sharing guidelines to our attention. We acknowledge the importance of making our author-generated code available to facilitate reproducibility and reuse.

We are committed to sharing the code supporting our findings and will make it publicly available upon manuscript publication. Meanwhile, we need more time to clean and document the code to meet the best practices outlined by PLOS ONE. We value your understanding and will ensure the code is prepared and shared without restrictions upon publication.

4. Please note that your Data Availability Statement is currently missing the direct link to access each database. If your manuscript is accepted for publication, you will be asked to provide these details on a very short timeline. We therefore suggest that you provide this information now, though we will not hold up the peer review process if you are unable.

Response:

Done! The DOI is 10.5281/zenodo.12699563

Response:

Thank you for your thorough review and for highlighting the importance of maintaining an accurate and up-to-date reference list. We have carefully reviewed our reference list to ensure that it is complete and correct. To verify the status of the cited papers, we searched the Retraction Watch Database and confirmed that none of the papers in our reference list have been retracted. As such, no changes were necessary to the reference list.

In addition, we have included new references related to the BERT model to support our analysis, as below:

 BertAA: BERT fine-tuning for Authorship Attribution

 Cross-domain authorship attribution using pre-trained language models

 Bert: Pre-training of deep bidirectional transformers for language understanding

 Unsupervised cross-lingual representation learning at scale

 Roberta: A robustly optimized bert pretraining approach

 Automated authorship attribution using advanced signal classification techniques

 A two level learning model for authorship authentication

Response to Reviewer #1 comments

General comments: This study presents a statistically robust investigation into the authorship attribution of the low-resource language Albabian through a strategic employment of multiclass classification techniques and machine learning approaches on chosen sets of linguistic features. The subsections of the article are neatly organized with an elaborated description of the analysis procedure and a clear presentation of experimental results. The insights derived from the results are also interesting. However, several points need to be addressed.

Response:

Thank you for your positive and encouraging feedback on our study. We are pleased that you found the organization, description of the analysis procedure, and presentation of the experimental results to be clear and insightful. We appreciate your recognition of our efforts and the robustness of our investigation into Albanian authorship attribution.

Regarding the specific points that need to be addressed, we have carefully considered each of your comments and have made the necessary revisions to the manuscript.

Comment 1: Clarification of Results in the Abstract Section: The authors are recommended to summarize concrete results in the abstract section, specifying the most suitable set of linguistic features for authorship attribution and the best machine learning approaches for Albanian texts, as proposed in the research questions.

Response:

Thank you for your valuable feedback. We agree that summarizing the concrete results in the abstract will enhance the scope and clarity of our study. In response, we have revised the abstract to include specific findings, highlighting the most effective set of linguistic features and the best machine-learning approaches for Albanian authorship attribution.

Comment 2.1: Possible Effect of “Average Number of Words (Tokens) per Author” on ML Algorithm Performance: In terms of data size for authorship attribution tasks, 10,000 words per author is considered the “reliable minimum for an authorial set” (Burrows, 2007). As presented in Table 2, “Authorship Statistics,” the average number of tokens per author in both genres is far too limited when measured against this standard.

Response:

Thank you for pointing out this important consideration. There seems to be a misunderstanding regarding the calculation of the average number of tokens per author. In our study, we calculated the average number of tokens per author per sample, not the total number of words per author. The formula used for calculating the average number of tokens per author (per sample) is as follows:

avg no of words (per sample)= (total no of words)/(total no of samples)

avg no of samples per author= (total no of samples)/(total no of authors)

avg no of tokens per author (per sample)= (avg no of words (per sample))/(total no of authors)

To address this and provide clarity, we have revised the dataset statistics table (Table 2) and included additional rows that provide the minimal total number of words per author, the maximal total number of words per author, and the average total number of words per author. With these updates, we believe in providing a clearer and more comprehensive overview of the dataset, addressing the concern regarding the minimum number of words per author.

Comment 2.2: The effect of factors such as “number of candidate authors” and “data size” has been fully acknowledged by studies such as Luyckx & Daelemans (2011). Could the performance of authorship attribution in the present study also be influenced by factors like “author set size”?

Response:

To address any potential concerns, we have revised our dataset statistics table to provide a clearer representation of the data distribution. This revision aims to indicate the sufficiency of our dataset for the analysis conducted. This clarification illustrates that our dataset meets the standards for a reliable authorial set, thereby mitigating concerns regarding the data size per author.

Regarding the “author set size,” we acknowledge that a larger number of candidate authors can increase the complexity of the classification task. We conducted our experiments with a carefully selected number of candidate authors, ensuring a diverse range of writing styles and genres, as well as maintaining a sufficient amount of data per author for effective analysis. However, we are also considering expanding the dataset and further investigating the impact of "author set size" in future work to provide a more comprehensive analysis of its effects on authorship attribution performance.

Response to Reviewer #2 comments

General comments: This paper introduced the extended Albanian authorship attribution corpus (A3C-e) and the most effective linguistic features and the most effective machine learning models. It took considerable efforts to build the Albanian authorship attribution corpus, consisting of the newsroom text and scanned books. Extensive experiments were also conducted to identify the most effective linguistic features and machine learning models. However, there is still some room for improvement.

Response:

Thank you for your positive feedback on our paper and for recognizing the efforts involved in building the extended Albanian authorship attribution corpus (A3C-e) and identifying the most effective linguistic features and machine learning methods. We appreciate your acknowledgment of the extensive experiments conducted in this study.

We understand that there is always room for improvement, and we have carefully considered your specific suggestions for enhancing the manuscript. Below, we address each of your comments and outline the revisions made to our work.

Comment 1: As an authorship attribution corpus, it’s better to include of a breakdown of samples by author. For example, how many samples are collected from one author? How are the samples distributed by authors?

Response:

As part of our authorship attribution corpus, we have included a detailed breakdown of samples by author (Figure 2) in the revised manuscript, specifying the number of samples collected from each author and how they are distributed.

Comment 2.1: It’s better to detail the imperative and reflective programming paradigms that were used to extract features.

Response:

To calculate the complete set of extracted features, we utilized a pipeline using the globals() function and the inspect module. Combining these two tools' functionality gave us a toolset for programmatically examining and manipulating objects in Python code.

The process involved executing the globals() function to obtain a dictionary of the current global symbol table and then using inspect to retrieve information about the user-defined functions that were available during the execution of the Python code.

By combining the functionality of these two tools, we inspected the current scope of the Python code and retrieved information about the user-defined functions that we manually programmed. A screenshot of a section of the code may be found below:

Comment 2.2: Are there any open-source tools used?

Response:

We want to clarify that all the features extracted for this study were carefully and manually engineered and programmed. We did not rely on any open-source tools for feature extraction. Instead, the custom scripts we developed ensure that the features are tailored specifically to the unique characteristics of the Albanian language and the requirements of the authorship attribution analysis.

Comment 2.3: Will the codes for feature extraction be released publicly or not?

Response:

We are committed to transparency and reproducibility and will make our code available upon the publication of our work.

Comment 3: For deep learning methods, BERT multilingual also support Albanian. It’s better to include BERT in the deep learning methods.

Response:

Thank you for your insightful suggestion. We appreciate the recommendation to include BERT-multilingual to provide a more comprehensive evaluation of deep learning methods for authorship attribution in Albanian.

BERT-multilingual and XLM-RoBERTA-base have been incorporated into our experimental setup. The manuscript now includes a comparison of its performance with other models (Table 8), demonstrating its effectiveness for Albanian texts.

Comment 4: It’s better to define F1, that is, the equation for F1 calculation.

Response:

Thank you for pointing this out. In response to this comment, we have added a definition of the F1 score, including the equation for its calculation, to the manuscript. This will ensure clarity and provide a precise understanding of how the F1 score is computed.

---

## [Editor Report · Decision Letter 1]

20 Aug 2024

Automatic Authorship Attribution in Albanian texts

PONE-D-24-28247R1

Dear Dr. Misini,

We’re pleased to inform you that your manuscript has been judged scientifically suitable for publication and will be formally accepted for publication once it meets all outstanding technical requirements.

Kind regards,

Muhammad Afzaal, PhD

Academic Editor

PLOS ONE
---

## [Editor Report · Acceptance letter]

27 Aug 2024

PONE-D-24-28247R1 

PLOS ONE

Dear Dr. Misini, 

I'm pleased to inform you that your manuscript has been deemed suitable for publication in PLOS ONE. Congratulations! Your manuscript is now being handed over to our production team.

Kind regards, 

on behalf of

Dr. Muhammad Afzaal 

Academic Editor

PLOS ONE